# Self-Training with Dynamic Weighting for Robust Gradual Domain Adaptation

**Zixi Wang**[*]
University of Electronic Science
and Technology of China
Chengdu, Sichuan, China

**Yushe Cao**[*]
Tsinghua University
Beijing, China

**Yubo Huang**[*]
Zhenguan AI Lab
Shenzhen, Guangdong, China
Southwest Jiaotong University
Chengdu, Sichuan, China

**Jinzhu Wei** [†]
Shanghai University
Shanghai, China
wbiozhu@gmail.com

**Jingzehua Xu** [‡]
Tsinghua University
Beijing, China
xjzh23@mails.tsinghua.edu.cn

**Shuai Zhang**
New Jersey Institute of Technology
Newark, NJ, United States

**Xin Lai**
Southwest Jiaotong University
Chengdu, Sichuan, China

## Abstract

In this paper, we propose a new method called *Self-Training with Dynamic Weighting* (STDW), which aims to enhance robustness in Gradual Domain Adaptation (GDA) by addressing the challenge of smooth knowledge migration from the source to the target domain. Traditional GDA methods mitigate domain shift through intermediate domains and self-training but often suffer from inefficient knowledge migration or incomplete intermediate data. Our approach introduces a dynamic weighting mechanism that adaptively balances the loss contributions of the source and target domains during training. Specifically, we design an optimization framework governed by a time-varying hyperparameter $\varrho$ (progressing from 0 to 1), which controls the strength of domain-specific learning and ensures stable adaptation. The method leverages self-training to generate pseudo-labels and optimizes a weighted objective function for iterative model updates, maintaining robustness across intermediate domains. Experiments on rotated MNIST, color-shifted MNIST, portrait datasets, and the Cover Type dataset demonstrate that STDW outperforms existing baselines. Ablation studies further validate the critical role of $\varrho$'s dynamic scheduling in achieving progressive adaptation, confirming its effectiveness in reducing domain bias and improving generalization. This work provides both theoretical insights and a practical framework for robust gradual domain adaptation, with potential applications in dynamic real-world scenarios. The code is available at https://github.com/Dramwig/STDW.

## 1  Introduction

---

[*]Equal contribution.

[†]Co-Corresponding author.

[‡]Corresponding author.

39th Conference on Neural Information Processing Systems (NeurIPS 2025).

The phenomenon of domain shift presents a fundamental challenge in machine learning systems, where the statistical properties of training (source) and deployment (target) domains exhibit significant divergence [1, 2]. As illustrated in Figure 1, this distributional mismatch frequently arises in practical applications where target domain annotations are either prohibitively expensive or fundamentally unavailable. Domain adaptation (DA) methodologies [3, 4] have emerged as a critical paradigm for addressing this challenge, particularly through the frame-

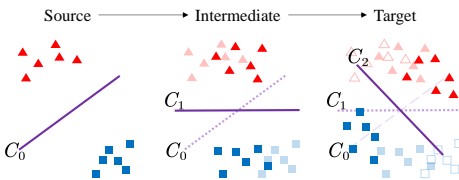

Figure 1: Overview of gradual domain adaptation. The classifiers perform accurately in classifying the sample points in the current domain and the neighboring domains, but fail when classifying the samples from the Source directly to the Target domain.

work of unsupervised domain adaptation (UDA) [5]. However, conventional UDA approaches demonstrate limited efficacy when confronted with substantial domain discrepancies or when transitional domain data is sparse or absent.

Recent theoretical and empirical advances have established Gradual Domain Adaptation (GDA) as a promising alternative for mitigating these limitations. The GDA framework, as formalized by [6] and extended by [7], implements a sequential knowledge migration mechanism that systematically reduces the inter-domain discrepancy through intermediate transitional phases. Although this paradigm demonstrates improved stability compared to direct adaptation approaches [8], current GDA implementations exhibit inefficient information flow between adjacent domains, resulting in gradual performance degradation throughout the adaptation trajectory [9]. Moreover, the sequential nature of GDA introduces compounding approximation errors, particularly when transitioning through numerous intermediate domains [10].

This paper proposed *Self-Training with Dynamic Weighting* (STDW), identify and address three critical challenges in existing GDA approaches: (1) suboptimal knowledge propagation across domains, (2) instability during transitional phases, and (3) limited generalization capability. Our STDW introduces a principled framework that fundamentally rethinks the domain adaptation paradigm through two key innovations:

- A theoretically-grounded optimization framework employing the time-dependent parameter $\varrho \in [0, 1]$, which orchestrates a smooth transition from source to target domain representation learning. This mechanism ensures controlled knowledge assimilation while maintaining model stability throughout the adaptation trajectory.

- An integrated self-training paradigm that synergistically combines pseudo-label refinement with adaptive loss reweighting, effectively addressing the domain shift problem through iterative model purification and confidence-aware sample selection.

Through extensive empirical validation across four benchmark datasets - including Rotated MNIST, Color-shifted MNIST, Portrait, and Cover Type datasets - we demonstrate that STDW achieves state-of-the-art performance, outperforming existing methods by significant margins. Comprehensive ablation studies further verify the critical role of our dynamic weighting mechanism and its superior ability to facilitate robust knowledge migration across domains. The proposed framework establishes new theoretical foundations for gradual domain adaptation while providing practical insights for real-world applications.

## 2 Related Work

**Domain Generalization (DG)** has emerged as a principled approach to address the fundamental challenge of distributional shift without requiring access to target domain data during training [11]. The theoretical underpinnings of DG trace back to kernel-based methods for learning domain-invariant representations, as formalized by [12], who established important connections between feature space alignment and generalization performance. Subsequent developments have significantly expanded this paradigm through adversarial learning frameworks that explicitly minimize the discriminability of domain-specific features while preserving task-relevant information [13]. A parallel line of research has demonstrated the effectiveness of data-centric approaches in simulating domain variations during training. The work of [14] pioneered sophisticated data augmentation strategies that generate synthetic

domain shifts through carefully designed transformations, effectively creating a continuum of virtual training environments.

More recently, the field has witnessed significant advances through the integration of meta-learning principles into the DG framework. [15] and [16] have reformulated the problem as a meta-optimization task, where models are trained to rapidly adapt to simulated domain shifts presented in episodic training scenarios. This meta-learning perspective provides formal generalization guarantees and has established new state-of-the-art performance across multiple benchmark datasets. While these approaches demonstrate impressive results in controlled evaluations, our analysis reveals persistent limitations when confronting substantial domain discrepancies, particularly in scenarios where the underlying data manifold undergoes complex nonlinear transformations. This observation motivates our investigation of gradual adaptation strategies that can better preserve model stability across significant distributional shifts.

**Gradual Domain Adaptation (GDA)** has emerged as a principled approach to address the fundamental challenge of substantial domain shifts by decomposing the adaptation process into a sequence of intermediate, more manageable steps. The theoretical foundations of GDA were established by [7] and [6] demonstrated that gradual transitions between domains can significantly improve adaptation performance compared to direct source-to-target approaches. This paradigm shift builds upon the key insight that complex real-world domain shifts often evolve gradually rather than occurring abruptly, suggesting that modeling the continuous transformation between domains may better capture the underlying data manifold structure.

Recent advances in GDA have focused on two primary directions: the generation of meaningful intermediate domains and the development of robust adaptation strategies across these domains. Theoretical work by [16] has provided rigorous mathematical frameworks for understanding the benefits of gradual adaptation, while algorithmic innovations have explored various approaches to constructing intermediate distributions. These include gradient flow-based geodesic paths [17] that preserve the intrinsic geometry of the data manifold, style-transfer interpolation techniques [10] that generate visually plausible intermediate samples, and optimal transport methods [9] that minimize the Wasserstein distance between consecutive domains. These methodological advancements collectively represent a significant step forward in addressing large domain gaps, though important challenges remain in scaling these approaches to high-dimensional spaces and ensuring their robustness to noisy or imperfect intermediate domains.

## 3 Problem Setup

**Domain Space** Let $\mathcal{Z} = \mathcal{X} \times \mathcal{Y}$ denote the measurable instance space, where $\mathcal{X} \subseteq \mathbb{R}^d$ is the $d$-dimensional input space and $\mathcal{Y} = \{1, 2, \ldots, k\}$ is the label space with $k$ denoting the number of classes.

**Gradually Shifting Domain** Following the gradual domain shift framework [6], we assume a sequence of $n + 1$ domains $\{\mathcal{D}_t\}_{t=0}^n$ defined over $\mathcal{Z}$. Here, $\mathcal{D}_0$ represents the labeled source domain, $\mathcal{D}_n$ denotes the unlabeled target domain, and the intermediate domains $\mathcal{D}_1, \ldots, \mathcal{D}_{n-1}$ form a smooth interpolation between them. Each $\mathcal{D}_t$ corresponds to a distinct data distribution, with the divergence between consecutive domains assumed to be sufficiently small to enable stable adaptation.

**Classification and Empirical Risk Minimization** The goal of supervised classification is to learn a hypothesis $f : \mathcal{X} \to \mathcal{Y}$ that minimizes the expected risk under a given distribution $\mathcal{D}_t$. Formally, given a loss function $l : \mathcal{Y} \times \mathcal{Y} \to \mathbb{R}_{\geq 0}$, the optimal parameters $\theta$ are obtained by solving:

$$\theta = \arg\min_\theta \mathbb{E}_{(x,y) \sim \mathcal{D}_t}[l(f(x;\theta), y)]. \tag{1}$$

**Domain Adaptation** In unsupervised domain adaptation (UDA), the model is trained on a labeled source domain $\mathcal{D}_\mathcal{S}$ but evaluated on an unlabeled target domain $\mathcal{D}_\mathcal{T}$, where $P_\mathcal{S}(x) \neq P_\mathcal{T}(x)$ while the conditional distribution $P(y \mid x)$ is assumed to be invariant. The central challenge lies in transferring knowledge across domains despite the distributional shift and the absence of labeled target data.

**Gradual Domain Adaptation** Gradual domain adaptation (GDA) addresses this challenge by leveraging a sequence of unlabeled intermediate domains $\{\mathcal{D}_t\}_{t=1}^{n-1}$ that bridge $\mathcal{D}_0$ and $\mathcal{D}_n$. Starting

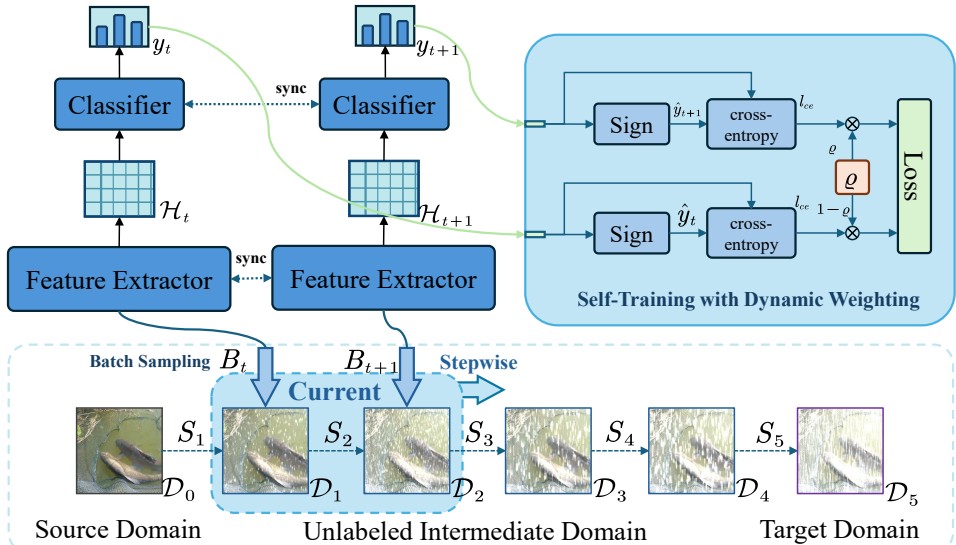

Figure 2: The framework of our *Self-Training with Dynamic Weighting* (STDW). It is designed to facilitate a smooth and controlled adaptation of a model from a source domain to a target domain. The hyperparameter $\varrho$ governs the trade-off between the two domains: when $\varrho = 0$, the model operates exclusively on the source domain, whereas as $\varrho$ increases toward 1, the model progressively shifts its emphasis toward the target domain. Pseudo-labels, generated by the model itself, guide the adaptation process, thereby ensuring a gradual and stable transition across domains.

from a model $f(\cdot; \theta_0)$ trained on the source domain $\mathcal{D}_0$, adaptation proceeds iteratively: at stage $t \in \{1, \ldots, n\}$, the current model $f(\cdot; \theta_{t-1})$ generates pseudo-labels for samples from $\mathcal{D}_t$, which are then used to update the model parameters via self-training:

$$\theta_t = \arg\min_{\theta'} \mathbb{E}_{x \sim \mathcal{D}_t} [l(f(x; \theta'), \hat{f}(x; \theta_{t-1}))]. \tag{2}$$

where $\hat{y}(x; \theta_{t-1}) = \arg\max_{y \in \mathcal{Y}} f_y(x; \theta_{t-1})$ denotes the hard pseudo-label assigned by the previous model. The resulting model $f(\cdot; \theta_t)$ is then employed to annotate data from the next domain $\mathcal{D}_{t+1}$. By constraining the distributional shift between consecutive stages to be small, this procedure ensures a controlled and stable propagation of knowledge from source to target, thereby mitigating error accumulation and enhancing generalization in the final target domain.

## 4 Methodology

This work proposes Self-Training with Dynamic Weighting (STDW), which reconstructs traditional self-training paradigms through a triple adaptive mechanism. Fig. 2 illustrates the overall framework of Self-Training with Dynamic Weighting (STDW).

### 4.1 Pseudo-labeling Dynamic Learning

Existing gradual domain adaptation (GDA) methodologies typically employ a static pseudo-labeling paradigm, wherein the entire unlabeled dataset $\mathcal{D} = \{x_1, \ldots, x_N\}$ is annotated in a single pass. This process involves a classifier $f(\cdot; \theta)$ assigning hard labels through:

$$\hat{f}(x) = \arg\max_y f(x)_y, \tag{3}$$

followed by discarding a fixed proportion (e.g., 10%) of low-confidence samples to mitigate noise [6]. However, residual mislabeling inevitably persists, leading to compounded errors across successive domains and exponential degradation in model accuracy—a phenomenon empirically validated in prior studies.

To address this limitation, we propose *dynamic pseudo-labeling*, a sequential strategy that alternates between incremental pseudo-label generation and classifier refinement. Specifically, we partition

$\mathcal{D}$ into $T$ disjoint mini-batches $\{B_1, \ldots, B_T\}$. At iteration $t$, the model $\theta_{t-1}$ labels samples within batch $B_t$ via:

$$\hat{y}_i^{(t)} = \arg\max_y f(x_i; \theta_{t-1})_y \quad \forall x_i \in B_t, \tag{4}$$

yielding pseudo-labels $\{\hat{y}_i^{(t)}\}$. These labels then drive a parameter update by minimizing the cross-entropy loss $\mathcal{L}$ over $B_t$:

$$\theta_t = \theta_{t-1} - \eta \nabla_\theta \mathcal{L}\left(f(B_t; \theta_{t-1}), \{\hat{y}_i^{(t)}\}\right), \tag{5}$$

where $\eta$ denotes the learning rate. This iterative process is formalized through a batch-wise update operator $\mathcal{U}$:

$$\theta_t = \mathcal{U}\left(\theta_{t-1}, B_t\right), \quad t = 1, \ldots, T. \tag{6}$$

By progressively refining pseudo-labels with an incrementally improved classifier, our approach circumvents the error accumulation inherent to static labeling schemes. This dynamic interaction between label generation and model adaptation ensures that each batch benefits from the enhanced discriminative capability acquired from preceding batches, thereby alleviating label noise propagation and fostering more stable domain adaptation. The resultant framework demonstrates superior robustness, as evidenced by our experimental analyses.

## 4.2 Cyclic Batch Matching Across Neighboring Domains

Data comes from a range of domains in the gradual domain adaptation scenario. Consider two neighboring domains $\mathcal{D}_l$ (left domain) and $\mathcal{D}_r$ (right domain), which are partitioned as $\mathcal{D}_l = \{B_{l,1}, B_{l,2}, \ldots, B_{l,n}\}, \mathcal{D}_r = \{B_{r,1}, B_{r,2}, \ldots, B_{r,m}\}$.

To facilitate robust inter-domain adaptation, we define cyclic sequences that allow batches from each domain to be matched in a fixed, periodic manner. Specifically, we define the sequences $\{B_l(t)\}_{t\geq 0}$ and $\{B_r(t)\}_{t\geq 0}$, with the mapping given by:

$$B_l(t) = B_{l,\,i(t)}, \quad B_r(t) = B_{r,\,j(t)}, \tag{7}$$

where the indexing functions $i(t)$ and $j(t)$ are defined as:

$$i(t) = (t \mod n) + 1,$$
$$j(t) = \left((t + \left\lfloor \frac{t}{n} \right\rfloor) \mod m\right) + 1. \tag{8}$$

This formulation ensures the batches are sampled cyclically over the two domains. The resulting cyclic matching can then be represented by the ordered pair sequence:

$$\left\{\left(B_{l,\,i(k)}, B_{r,\,j(k)}\right)\right\}_{k\geq 0}. \tag{9}$$

Consider a sequence of $n+1$ domains $\{\mathcal{D}_t\}_{t=0}^n$, where each domain $\mathcal{D}_t$ consists of $m$ data batches denoted by $\{B_{t,1}, B_{t,2}, \ldots, B_{t,m(t)}\}$. We define a time-varying classifier $f^{(t,k)}$ to be the model after the $k$-th batch update within domain $\mathcal{D}_{t-1}$ and $\mathcal{D}_t$. Its evolution is governed by the incremental update rule:

$$\theta^{(t,k+1)} = \Phi\left(\theta^{(t,k)}, B_{t-1,i(k)}, B_{t,j(k)}\right), \tag{10}$$

where $\Phi$ is an optimization operator defined via the following objective:

$$\Phi = \arg\min_{\theta'} \left\{\mathbb{E}_{x \sim B_{t,i(k)}}\left[\ell_{\text{ce}}\left((f(x_i; \theta'), \hat{f}(x; \theta))\right)\right] + \mathbb{E}_{x \sim B_{t,j(k)}}\left[\ell_{\text{ce}}\left((f(x_i; \theta'), \hat{f}(x; \theta))\right)\right]\right\}, \tag{11}$$

Just like Eq. 5 and Eq. 6, we can update $\theta$ through $\Phi$.

## 4.3 Stepwise Dynamic Osmosis

We formulate the cross-domain adaptation process as the following optimization problem:

$$\theta^{(t,k+1)} = \Phi\left(\theta^{(t,k)}, B_{t-1,i(k)}, B_{t,j(k)}, \varrho\right), \tag{12}$$

$$\Phi = \arg\min_{\theta'} \left\{ (1-\varrho)\mathbb{E}_{x \sim B_{t,i(k)}} \left[ \ell_{\text{ce}}\Big( (f(x_i; \theta'), \hat{f}(x; \theta) \Big) \right] + \varrho \mathbb{E}_{x \sim B_{t,j(k)}} \left[ \ell_{\text{ce}}\Big( (f(x_i; \theta'), \hat{f}(x; \theta) \Big) \right] \right\}.$$

(13)

where $\varrho \in [0, 1]$ is a hyperparameter that balances the contribution of the current domain $\mathcal{D}_i$ and the subsequent domain $\mathcal{D}_{i+1}$. As the training progresses, $\varrho$ increases from 0 to 1, thereby gradually migrating the learned knowledge from the source domain $\mathcal{D}_i$ to the target domain $\mathcal{D}_{i+1}$.

---

**Algorithm 1** Self-Training with Dynamic Weighting (STDW)

---

1: **Input:** Source batches $\{B_{0,k}\}_{k=1}^m$, domain sequence $\{\mathcal{D}_t\}_{t=1}^n$, initial model $f^{(0,0)}$, Inter-domain migration steps: $s$
2: **Output:** Adapted model $f^{(n,m)}$
3: $g \leftarrow 1/s$
4: **for** $t = 1$ TO $n$ **do**
5:     **while** $\varrho \leq 1$ **do**
6:         **for** $k = 1$ TO $m \cdot epochs$ **do**
7:             Fetch batch $B_{t,k}$, generate pseudo-labels: $\hat{y} \leftarrow \hat{f}(x; \theta^{(t,k-1)})$
8:             Use Eq. 12, Eq. 13 update model: $\theta^{(t,k)} \leftarrow \Phi(\theta^{(t,k-1)}, B_{t-1,i(k)}, B_{t,j(k)}, \varrho)$
9:         **end for**
10:         $\varrho \leftarrow \varrho + g$
11:     **end while**
12:     Switch to the next domain: $\theta^{(t+1,0)} \leftarrow \theta^{(t,m)}$
13: **end for**

---

## 5 Experiments

### 5.1 Environments Setup

**Datasets.** Following established gradual domain adaptation protocols [6, 9], we utilize four core datasets: Rotated MNIST [18] and Color-Shift MNIST for controlled synthetic transformations, Portraits Dataset [19] for real-world temporal shifts, and Cover Type Dataset [20] for tabular domain adaptation. The Rotated MNIST dataset, derived from the standard MNIST digits [18], introduces a progressive geometric transformation where source images (0° rotation) gradually evolve to target images (45° rotation) through intermediate domains of increasing rotation angles [9]. Similarly, Color-Shift MNIST examines photometric transformations by systematically varying pixel intensity ranges from [0,1] in the source domain to [1,2] in the target domain [9]. For real-world evaluation, the Portraits Dataset [19] provides a temporal domain shift spanning over a century (1905-2013), partitioned into nine chronological domains with distinct fashion and photographic characteristics [6]. The Cover Type Dataset [20] offers a challenging tabular scenario where domains are constructed based on ecological proximity to water sources [6]. To further stress-test our method under severe distribution shifts, we incorporate two additional corruption benchmarks: CIFAR-10-C and CIFAR-100-C [21], which systematically apply 15 types of image corruptions at varying severity levels, treating the highest severity as the target domain.

**Implementation.** For image datasets (Rotated MNIST, Color-Shift MNIST, Portraits), we implement a convolutional neural network comprising three 32-channel convolutional layers followed by two 256-unit fully connected layers. The tabular Cover Type dataset utilizes a progressively expanding fully connected architecture (128-256-512 units). All models incorporate ReLU activations, batch normalization [22], and dropout regularization [23], optimized using Adam [24] with carefully tuned hyperparameters. The transport network architecture features residual-connected generators and three-layer discriminators (128 hidden units per layer) to facilitate domain transitions. Our evaluation systematically varies the number of intermediate domains (0-4) to analyze adaptation trajectory smoothness. For the corruption benchmarks, we employ established robust architectures: WideResNet-28 [25] for CIFAR-10-C and ResNeXt-29 [26] for CIFAR-100-C, following Robust-Bench protocols [27, 28] to ensure comparable evaluation conditions. All experiments are conducted on NVIDIA RTX 4090 GPUs with identical random seeds for reproducibility.

**Benchmarks.** Our comparative analysis encompasses the following state-of-the-art domain adaptation methods spanning both conventional and gradual adaptation paradigms. The evaluation includes three unsupervised domain adaptation baselines: DANN [29] (adversarial adaptation), DeepCoral [30] (correlation alignment), and DeepJDOT [31] (joint distribution optimal transport). For gradual domain adaptation, we benchmark against five recent approaches: GST [6] (gradual self-training), IDOL [8] (intermediate domain learning), GOAT [9] (geodesic optimal transport), GGF [17] (gradual geometric flow), and GNF [32] (gradual normalizing flows). To evaluate the adaptability of the model in a slowly evolving environment, we conducted experiments using TTA application methods including TENT-continual [33], AdaContrast [34], CoTTA [35], GTTA-MIX [36] and the GDA method GST. In all the experiments, we took the error rate as the main indicator in order to make a direct comparison with the existing literature and provide a clear explanation of the model performance under different adaptation challenges at the same time.

## 5.2 Results

Table 1 presents a comprehensive comparison of our method against state-of-the-art unsupervised domain adaptation (UDA) and gradual domain adaptation (GDA) approaches across multiple benchmark datasets. Our *Self-Training with Dynamic Weighting* (STDW) consistently achieves the best performance, establishing new state-of-the-art results with substantial margins. Specifically, STDW improves absolute accuracy by **11.2%**, **6.5%**, **0.94%**, and **4.3%** over the second-best methods. These gains underscore the effectiveness of STDW in modeling continuous distributional shifts through its dynamic weighting mechanism, in contrast to static alignment strategies (e.g., DANN) or single-step gradual adaptation schemes. Moreover, among the limited set of domain adaptation frameworks capable of operating across diverse adaptation scenarios—including both abrupt and gradual shifts—STDW demonstrates exceptional versatility and robustness, highlighting its broad applicability.

Table 1: Benchmarks Comparison on different datasets, including UDA methods and GDA methods.

| UDA/GDA methods | Rotated MNIST | Color-Shift MNIST | Portraits | Cover Type |
|---|---|---|---|---|
| DANN [37] | 44.2 | 56.5 | 73.8 | 65.2 |
| DeepCoral [30] | 49.6 | 63.5 | 71.9 | 66.8 |
| DeepJDOT [31] | 51.6 | 65.8 | 72.5 | 67.4 |
| GST [6] | 83.8 | 74.0 | 82.6 | 73.5 |
| IDOL [8] | 87.5 | 78.3 | 85.5 | 72.1 |
| GOAT [9] | 86.4 | 91.8 | 83.6 | 69.9 |
| GGF [17] | 67.72 | 73.6 | 86.1 | 71.2 |
| CNF [38] | 62.55 | 70.4 | 84.6 | 70.8 |
| STDW (Ours) | **97.6** | **98.3** | **87.1** | **74.2** |

The experimental results presented in Table 3 demonstrate the effectiveness of our proposed STDW method compared to seven state-of-the-art approaches across 15 distinct corruption types at severity level 5 achieving superior performance overall, attaining the lowest mean error rate of 15.5% on CIFAR-10-C and 25.8% on CIFAR-100-C. This represents a significant improvement over the source-only baseline (43.5% and 46.4% respectively) and outperforms all comparison methods, including the closest competitor GTTA-MIX (15.6% and 28.9%). Notably, STDW achieves the best performance on 11 out of 15 corruption types for CIFAR-10-C and 14 out of 15 for CIFAR-100-C, demonstrating remarkable consistency across diverse corruption patterns.

Furthermore, for high-frequency corruptions like Gaussian noise, shot noise, and impulse noise, STDW shows particularly strong performance with error rate reductions of 3.3-11.6 percentage points compared to the next best method. This suggests our approach effectively handles high-frequency distortions that typically challenge conventional adaptation methods. For geometric distortions (e.g., zoom, motion) and weather-based corruptions (e.g., snow, frost), STDW maintains consistent advantages, indicating robust performance across both spatial and photometric transformations. The consistent superiority of STDW across both datasets and corruption types validates our key contributions: (1) the effectiveness of progressive domain transport through weighted intermediate steps, and (2) the robustness of our approach to diverse types of domain shifts.

Table 2: Classification error rate (%) on the highest damage severity level 5 after gradual adaptation of CIFAR100-to-CIFAR100C with progressively higher damage. For all method results evaluated on the same ResNeXt-29 that has been trained on the source domain, for STDW we fixed the use of 2 intermediate steps. We report the average performance of our method over 5 runs. Red indicates the best value, Green indicates the second-best.

| | Methods | Source only | BN-1 | TENT-cont. | AdaContrast | CoTTA | GTTA-MIX | GST | STDW (ours) |
|---|---|---|---|---|---|---|---|---|---|
| | Gradual | ✗ | ✗ | ✗ | ✗ | ✗ | ✗ | ✓ | ✓ |
| CIFAR-10-C | Gaussian | 72.3 | 28.1 | 24.8 | 29.1 | 24.3 | 23.4 | 50.0 | 21.5 |
| | Shot | 65.7 | 26.1 | 20.6 | 22.5 | 21.3 | 18.3 | 43.9 | 19.0 |
| | Impulse | 72.9 | 36.3 | 28.6 | 30.0 | 26.6 | 25.5 | 50.3 | 27.9 |
| | Defocus | 46.9 | 12.8 | 14.4 | 14.0 | 11.6 | 10.1 | 20.6 | 9.6 |
| | Glass | 54.3 | 35.3 | 31.1 | 32.7 | 27.6 | 27.3 | 51.2 | 28.3 |
| | Motion | 34.8 | 14.2 | 16.5 | 14.1 | 12.2 | 11.6 | 17.2 | 10.3 |
| | Zoom | 42.0 | 12.1 | 14.1 | 12.0 | 10.3 | 10.1 | 16.7 | 8.5 |
| | Snow | 25.1 | 17.3 | 19.1 | 16.6 | 14.8 | 14.1 | 17.5 | 3.0 |
| | Frost | 41.3 | 17.4 | 18.6 | 14.9 | 14.1 | 13.0 | 24.3 | 13.3 |
| | Fog | 26.0 | 15.3 | 18.6 | 14.4 | 12.4 | 10.9 | 17.5 | 11.7 |
| | Bright | 9.3 | 8.4 | 12.2 | 8.1 | 7.5 | 7.4 | 6.9 | 7.1 |
| | Contrast | 46.7 | 12.6 | 20.3 | 10.0 | 10.6 | 9.0 | 13.2 | 8.8 |
| | Elastic | 26.6 | 23.8 | 25.7 | 21.9 | 18.3 | 19.4 | 24.9 | 19.6 |
| | Pixelate | 58.5 | 19.7 | 20.8 | 17.7 | 13.4 | 14.5 | 39.9 | 13.4 |
| | JPEG | 30.3 | 27.3 | 24.9 | 20.0 | 17.3 | 19.8 | 26.6 | 20.1 |
| | Mean | 43.5 | 20.4 | 20.7 | 18.5 | 16.2 | 15.6 | 28.1 | 15.5 |
| CIFAR-100-C | Gaussian | 73.0 | 42.1 | 37.2 | 42.3 | 40.1 | 36.4 | 49.8 | 31.0 |
| | Shot | 68.0 | 40.7 | 35.8 | 36.8 | 37.7 | 32.1 | 56.7 | 27.6 |
| | Impulse | 39.4 | 42.7 | 41.7 | 38.6 | 39.7 | 34.0 | 32.3 | 26.1 |
| | Defocus | 29.3 | 27.6 | 37.9 | 27.7 | 26.9 | 24.4 | 22.5 | 22.1 |
| | Glass | 54.1 | 41.9 | 51.2 | 40.1 | 38.0 | 35.2 | 41.6 | 31.6 |
| | Motion | 30.8 | 29.7 | 48.3 | 29.1 | 27.9 | 25.9 | 25.0 | 22.9 |
| | Zoom | 28.8 | 27.9 | 48.5 | 27.5 | 26.4 | 23.9 | 23.3 | 22.6 |
| | Snow | 39.5 | 34.9 | 58.4 | 32.9 | 32.8 | 28.9 | 30.3 | 25.0 |
| | Frost | 45.8 | 35.0 | 63.7 | 30.7 | 31.8 | 27.5 | 32.2 | 25.4 |
| | Fog | 50.3 | 41.5 | 71.1 | 38.2 | 40.3 | 30.9 | 38.1 | 25.8 |
| | Bright | 29.5 | 26.5 | 70.4 | 25.9 | 24.7 | 22.6 | 22.1 | 21.9 |
| | Contrast | 55.1 | 30.3 | 82.3 | 28.3 | 26.9 | 23.4 | 27.0 | 24.3 |
| | Elastic | 37.2 | 35.7 | 88.0 | 33.9 | 32.5 | 29.4 | 33.1 | 27.2 |
| | Pixelate | 74.7 | 32.9 | 88.5 | 33.3 | 28.3 | 25.5 | 40.8 | 22.9 |
| | JPEG | 41.2 | 41.2 | 90.4 | 36.2 | 33.5 | 33.3 | 35.8 | 30.0 |
| | Mean | 46.4 | 35.4 | 60.9 | 33.4 | 32.5 | 28.9 | 33.3 | 25.8 |

Table 3: Comparison of accuracy for our STDW method across various datasets, considering different numbers of given intermediate domains (including the source and target domains). Results are averaged over 5 runs, with error bars representing the 95% confidence interval of the mean.

| | Rotated MNIST | | | | | Color-Shift MNIST | | | | |
|---|---|---|---|---|---|---|---|---|---|---|
| # Given | # Inter-domain counts in STDW | | | | | # Inter-domain counts in STDW | | | | |
| Domains | 0 | 1 | 2 | 3 | 4 | 0 | 1 | 2 | 3 | 4 |
| 2 | 81.5±1.3 | 82.8±3.5 | 81.7±4.4 | 82.0±6.4 | **83.6±1.2** | 86.2±0.1 | **96.5±0.0** | 87.2±3.6 | 86.6±0.0 | 86.6±0.1 |
| 3 | 95.9±0.4 | 96.5±0.8 | 96.8±0.3 | **96.9±0.1** | 96.8±0.1 | 98.1±0.0 | 98.2±0.0 | 98.2±0.0 | **98.3±0.0** | **98.3±0.0** |
| 4 | 96.5±0.2 | **96.9±0.1** | 96.8±0.0 | 96.8±0.0 | 96.8±0.1 | **98.3±0.0** | **98.3±0.0** | **98.3±0.0** | **98.3±0.0** | **98.3±0.0** |
| 5 | 96.8±0.1 | 96.9±0.1 | 96.7±0.1 | 96.9±0.0 | **97.0±0.1** | **98.3±0.0** | **98.3±0.0** | **98.3±0.0** | 98.2±0.0 | **98.3±0.0** |
| 6 | 96.9±0.1 | 96.9±0.1 | 96.9±0.1 | **97.6±0.0** | 96.9±0.0 | **98.3±0.0** | **98.3±0.0** | **98.3±0.0** | 98.2±0.0 | 98.2±0.0 |

| | Portraits | | | | | Cover Type | | | | |
|---|---|---|---|---|---|---|---|---|---|---|
| # Given | # Inter-domain counts in STDW | | | | | # Inter-domain counts in STDW | | | | |
| Domains | 0 | 1 | 2 | 3 | 4 | 0 | 1 | 2 | 3 | 4 |
| 2 | 83.7±0.3 | 84.4±0.8 | **85.1±0.6** | **85.1±0.8** | **85.1±0.5** | 69.7±0.0 | 70.0±0.0 | 70.5±0.1 | 71.2±0.0 | **71.8±0.1** |
| 3 | 84.1±0.3 | 84.6±0.2 | **84.8±0.5** | 84.7±0.1 | 84.7±0.2 | 70.1±0.0 | 72.3±0.0 | 73.9±0.0 | 74.3±0.0 | **74.4±0.1** |
| 4 | 83.8±0.7 | **84.0±0.1** | **84.0±0.2** | 83.8±0.3 | 83.9±0.1 | 71.5±0.0 | 73.8±0.0 | **74.2±0.0** | 74.0±0.0 | **74.2±0.0** |
| 5 | 84.8±0.4 | 84.8±0.4 | 84.8±0.4 | 84.8±0.2 | **84.9±0.4** | 72.5±0.1 | **74.2±0.0** | 74.1±0.0 | **74.2±0.1** | 73.9±0.0 |
| 6 | 85.3±0.9 | **86.1±0.4** | 85.8±1.0 | 85.6±1.8 | 85.4±0.9 | 73.1±0.0 | **74.1±0.0** | 73.4±0.0 | 73.9±0.0 | 73.4±0.0 |

We present a comparative evaluation of the proposed STDW framework across four benchmark datasets: *Rotated MNIST*, *Color-Shift MNIST*, *Portraits*, and *Cover Type*, with detailed results provided in Table 3. For each dataset, experiments were repeated multiple times under identical conditions, and the reported metrics represent the mean performance together with their associated variance intervals (e.g., $\pm$ one standard deviation). The leftmost column in each table corresponds to the baseline obtained via adversarial training alone—specifically, a domain-adversarial setup without the integration of flow matching—serving as a reference point to quantify the contribution of our proposed components.

The experimental results demonstrate the effectiveness of the STDW framework across the four benchmark datasets. For each dataset, we systematically vary two key factors: (i) the total number of available domains (including the source and target), and (ii) the number of inter-domain adaptation steps employed in the STDW procedure. We evaluate model performance as the number of adaptation steps increases, thereby providing a fine-grained analysis of how the granularity of domain interpolation influences the overall efficacy of the adaptation process. Notably, the observed standard deviations across five independent runs remain consistently low (all $\leq 1.2\%$), underscoring the stability and reliability of our method under diverse domain configurations.

## 5.3 Ablation Study

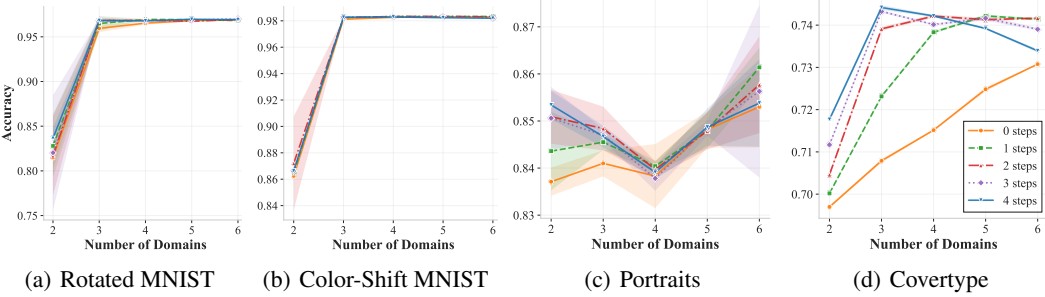

(a) Rotated MNIST     (b) Color-Shift MNIST     (c) Portraits     (d) Covertype

Figure 3: Ablation of STDW under different domain configurations and step counts. Accuracy on four benchmarks when varying the total number of domains (x-axis) and the inter-domain step counts used by STDW (curves). Each curve labeled "0–4 steps" indicates how many stepwise dynamic osmosis updates are performed per adjacent domain pair.

To investigate the impact of intermediate domains on adaptation performance, we conduct ablation studies on four benchmark datasets under a non-STDW baseline, varying the number of domains from 2 (i.e., source and target only) to 6 (i.e., source, target, and four unlabeled intermediate domains). The results, summarized in Figure 3, reveal a consistent trend: the inclusion of intermediate domains leads to substantial improvements in classification accuracy across most settings, accompanied by a notable reduction in standard deviation (computed over multiple runs). These findings strongly support the hypothesis that gradual, stepwise domain adaptation—enabled by intermediate distributions—plays a pivotal role in enhancing both the effectiveness and stability of domain generalization, particularly under significant source–target shifts.

Table 4: Comparison of STDW Method (Ours) with baselines across different domain adaptation steps on Rotated MNIST and Portraits datasets.

| Method | Rotated MNIST | | | | | Portraits | | | | |
|---|---|---|---|---|---|---|---|---|---|---|
| | 0 | 1 | 2 | 3 | 4 | 0 | 1 | 2 | 3 | 4 |
| Ours | $83.3 \pm 0.9$ | $85.0 \pm 0.5$ | $\mathbf{86.1 \pm 0.4}$ | $\mathbf{86.9 \pm 0.2}$ | $\mathbf{88.1 \pm 1.5}$ | $82.9 \pm 1.2$ | $\mathbf{84.6 \pm 0.2}$ | $\mathbf{85.0 \pm 0.9}$ | $\mathbf{85.1 \pm 0.2}$ | $\mathbf{85.3 \pm 0.1}$ |
| Sorted | $\mathbf{84.1 \pm 0.8}$ | $\mathbf{86.4 \pm 0.6}$ | $86.0 \pm 1.7$ | $86.3 \pm 0.1$ | $85.7 \pm 0.5$ | $82.7 \pm 0.5$ | $84.0 \pm 0.6$ | $84.2 \pm 0.1$ | $84.3 \pm 0.2$ | $84.5 \pm 0.1$ |
| Rand | $83.4 \pm 0.2$ | $80.9 \pm 7.0$ | $84.5 \pm 2.8$ | $86.3 \pm 0.9$ | $86.1 \pm 0.4$ | $82.4 \pm 0.5$ | $84.0 \pm 0.6$ | $84.2 \pm 0.2$ | $84.3 \pm 0.2$ | $84.1 \pm 0.1$ |
| Fixed | $83.3 \pm 0.0$ | $83.7 \pm 0.0$ | $83.8 \pm 0.1$ | $83.8 \pm 0.1$ | $84.1 \pm 0.0$ | $\mathbf{83.9 \pm 0.3}$ | $83.4 \pm 0.0$ | $83.1 \pm 2.1$ | $84.1 \pm 0.1$ | $84.7 \pm 0.3$ |

Additionally, we compare our stepwise dynamic osmosis strategy—referred to as *Ours (Equal)*, which linearly increases the weighting hyperparameter $\varrho$ in equal increments across adaptation steps—against three representative baselines: *Fixed* ($\varrho = 0.5$ held constant), *Rand* ($\varrho$ sampled independently from a uniform distribution $\mathcal{U}(0, 1)$ at each step), and *Sorted* (a variant of *Rand* where

sampled $\varrho$ values are sorted in ascending order to enforce monotonicity). Experiments are conducted on the *Rotated MNIST* and *Portraits* datasets under a two-domain setting (source and target only), with the number of inter-domain adaptation steps varied from 0 to 4. Results, reported in Table 4, show that *Equal* consistently achieves the highest final accuracy—for instance, attaining 88.1% on *Rotated MNIST* at Step 4, compared to 84.1% and 86.1% for *Fixed* and *Rand*, respectively. Moreover, *Equal* exhibits significantly lower variance across runs (e.g., $\pm 1.5$ vs. $\pm 7.0$ at Step 1 on *Rotated MNIST*), highlighting its superior stability. These results demonstrate that a controlled, monotonic increase of $\varrho$—as opposed to static, random, or even sorted-random schedules—yields more reliable and optimal adaptation performance.

# 6   Conclustion

This paper presents Self-Training with Dynamic Weighting (STDW), an innovative approach to Gradual Domain Adaptation (GDA) that systematically addresses the fundamental challenge of smooth knowledge migration across evolving domains. The proposed methodology introduces a dynamic weighting mechanism governed by the hyperparameter $\varrho$, which continuously modulates the relative influence between source and target domains throughout the adaptation process. This dynamic balancing enables a progressive and efficient knowledge migration within the self-training framework, effectively mitigating domain bias while maintaining model stability. Comprehensive experimental validation across multiple benchmark datasets confirms that STDW achieves superior performance compared to existing state-of-the-art methods, demonstrating significant improvements in both classification accuracy and model robustness under varying domain shift conditions.

# Acknowledgments

The work of Shuai Zhang was supported by National Science Foundation (NSF) #2349879.

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

## Limitations

While the proposed STDW method demonstrates significant improvements in gradual domain adaptation, several limitations should be acknowledged. First, the performance of STDW relies heavily on the availability and quality of intermediate domains, which may not always be accessible or well-defined in real-world scenarios. Second, the dynamic weighting mechanism, though effective, introduces additional hyperparameters (e.g., the scheduling of $\varrho$) that require careful tuning, potentially increasing the computational overhead during training. Third, the method assumes a smooth transition between domains, which might not hold for highly discontinuous or abrupt domain shifts. Additionally, the current experiments focus primarily on image and tabular data, leaving its applicability to other data modalities (e.g., text or time-series) unexplored. Finally, the theoretical analysis, while rigorous, relies on assumptions such as bounded Wasserstein distances and Lipschitz continuity, which may not fully capture the complexity of all practical domain adaptation problems. These limitations highlight directions for future work, such as automating hyperparameter selection and extending the framework to handle more diverse and challenging domain shifts.

## A   Supplementary Experimental Results

Classification correctness (%) for the CIFAR10-to-CIFAR-10-C, CIFAR100-to-CIFAR-100-C, and ImageNet-to-ImageNet-C adaptation task on the highest corruption severity level 5. For CIFAR-10-C the results are evaluated on WideResNet-28, for CIFAR-100-C on ResNeXt-29, and for Imagenet-C, ResNet-50 is used.

Table 5: Classification correctness (%) on the highest damage severity level 5 after gradual adaptation of CIFAR10-to-CIFAR10C with progressively higher damage. For all method results evaluated on the same WideResNet-28 that has been trained on the source domain, for STDW we fixed the use of 2 intermediate steps. We report the average performance of our method over 5 runs.

| Method | # Given Domains | gaussian | shot | impulse | defocus | glass | motion | zoom | snow | frost | fog | brightness | contrast | elastic | pixelate | jpeg | Mean |
|---|---|---|---|---|---|---|---|---|---|---|---|---|---|---|---|---|---|
| Source | - | 27.7 | 34.3 | 27.1 | 53.1 | 45.7 | 65.2 | 58.0 | 74.9 | 58.7 | 74.0 | 90.7 | 53.3 | 73.4 | 41.5 | 69.7 | 56.5 |
| GST | 2 | 37.0 | 40.9 | 33.8 | 57.5 | 46.3 | 67.1 | 58.8 | 76.7 | 60.6 | 76.9 | 91.9 | 60.6 | 74.1 | 44.8 | 70.5 | 59.8 |
| | 3 | 38.3 | 44.0 | 39.1 | 73.8 | 46.5 | 70.1 | 68.1 | 80.2 | 65.1 | 79.4 | 92.6 | 80.4 | 71.5 | 46.4 | 71.2 | 64.5 |
| | 4 | 41.4 | 48.7 | 41.5 | 76.2 | 48.2 | 76.8 | 75.7 | 80.0 | 71.8 | 81.5 | 93.0 | 85.4 | 75.0 | 51.9 | 72.1 | 68.0 |
| | 5 | 42.5 | 50.3 | 45.2 | 79.2 | 49.1 | 78.0 | 78.5 | 80.4 | 72.2 | 82.0 | **93.1** | 86.1 | 75.0 | 54.1 | 73.1 | 69.2 |
| | 6 | 50.0 | 56.1 | 49.7 | 79.4 | 48.8 | 82.8 | 83.3 | 82.5 | 75.7 | 82.5 | **93.1** | 86.8 | 75.1 | 60.1 | 73.4 | 71.9 |
| GOAT | 2 | 27.7 | 34.3 | 27.1 | 53.1 | 45.8 | 65.3 | 58.1 | 74.9 | 58.7 | 74.0 | 90.7 | 53.4 | 73.5 | 41.6 | 69.8 | 56.5 |
| | 3 | 27.5 | 34.3 | 27.0 | 53.2 | 45.8 | 65.4 | 58.2 | 74.9 | 58.8 | 74.0 | 90.7 | 53.5 | 73.4 | 41.7 | 69.8 | 56.5 |
| | 4 | 27.4 | 34.2 | 26.9 | 53.2 | 45.6 | 58.2 | 65.4 | 75.0 | 59.0 | 74.0 | 90.6 | 53.4 | 73.6 | 41.8 | 69.8 | 56.5 |
| | 5 | 27.3 | 34.2 | 26.9 | 53.3 | 45.6 | 65.5 | 58.4 | 75.1 | 59.0 | 74.0 | 90.7 | 53.4 | 73.6 | 41.8 | 69.8 | 56.6 |
| | 6 | 27.3 | 34.3 | 27.0 | 53.3 | 45.5 | 65.7 | 58.5 | 75.1 | 59.0 | 74.0 | 90.7 | 53.4 | 73.6 | 41.9 | 69.8 | 56.6 |
| STDW (Ours) | 2 | 73.6 | 75.5 | 65.4 | 88.5 | 66.4 | 87.5 | 89.2 | 84.0 | 83.5 | 86.2 | 92.5 | 88.8 | 77.9 | 82.2 | 74.0 | 81.0 |
| | 3 | 75.0 | 77.0 | 68.3 | 89.5 | 69.2 | 88.5 | 90.7 | 85.2 | 84.7 | 87.5 | 92.9 | 90.3 | 79.1 | 83.9 | 76.4 | 82.5 |
| | 4 | 76.4 | 80.0 | 69.6 | 90.0 | 69.7 | 88.9 | 91.4 | 86.1 | 85.6 | 88.2 | 92.9 | 90.9 | 79.7 | 85.2 | 77.5 | 83.5 |
| | 5 | 77.2 | 80.0 | 70.9 | 90.4 | 70.9 | 89.4 | **91.5** | 86.6 | 86.3 | 88.3 | 92.8 | **91.2** | 80.1 | 86.2 | 78.7 | 84.0 |
| | 6 | **78.5** | **81.0** | **72.1** | 90.4 | **71.7** | 89.7 | 91.4 | **97.0** | **86.7** | **88.3** | 92.7 | 90.8 | **80.4** | **86.6** | **79.9** | **84.5** |

## B   Theoretical Arguments

### B.1   Lyapunov Stability Under Cyclic Matching and Switching Anchors

For completeness, we extend the stability argument to the piecewise-smooth path constructed for discontinuous shifts. Let $\theta_t$ denote the parameters at step $t$ and consider the mixed loss on a matched pair of mini-batches

$$L_d(\theta_t) = \varrho_t \ell\big(\theta_t, \mathcal{B}_t^{(d)}\big) + (1 - \varrho_t)\ell\big(\theta_t, \mathcal{B}_t^{(d-1)}\big), \tag{14}$$

Table 6: Classification correctness (%) on the highest damage severity level 5 after gradual adaptation of CIFAR100-to-CIFAR100C with progressively higher damage. For all method results evaluated on the same ResNeXt-29 that has been trained on the source domain, for STDW we fixed the use of 2 intermediate steps. We report the average performance of our method over 5 runs.

| Method | #Given Domains | gaussian | shot | impulse | defocus | glass | motion | zoom | snow | frost | fog | brightness | contrast | elastic | pixelate | jpeg | Mean |
|---|---|---|---|---|---|---|---|---|---|---|---|---|---|---|---|---|---|
| Source | - | 27.0 | 12.0 | 60.6 | 70.7 | 45.9 | 69.2 | 71.2 | 60.5 | 54.2 | 49.7 | 70.5 | 44.9 | 62.8 | 25.3 | 58.8 | 53.6 |
| GST | 2 | 49.4 | 52.3 | 62.4 | 72.4 | 51.9 | 70.4 | 72.6 | 64.7 | 63.0 | 56.5 | 73.7 | 61.0 | 64.0 | 50.2 | 60.0 | 61.6 |
| | 3 | 48.3 | 50.4 | 62.8 | 74.2 | 54.7 | 71.8 | 74.4 | 66.6 | 63.8 | 57.4 | 74.9 | 64.0 | 64.1 | 49.4 | 61.1 | 62.5 |
| | 4 | 47.7 | 49.5 | 64.2 | 75.6 | 56.1 | 73.0 | 75.7 | 67.3 | 65.6 | 59.1 | 76.5 | 67.2 | 65.5 | 50.3 | 61.9 | 63.7 |
| | 5 | 50.0 | 52.0 | 66.2 | 76.6 | 57.9 | 73.8 | 76.2 | 68.1 | 66.8 | 60.3 | 77.2 | 69.1 | 66.5 | 53.7 | 63.4 | 65.2 |
| | 6 | 50.2 | 53.3 | 67.7 | 77.5 | 58.6 | 75.0 | 76.7 | 69.7 | 67.8 | 61.9 | 77.9 | 73.0 | 66.9 | 59.2 | 64.2 | 66.7 |
| GOAT | 2 | 27.1 | 32.0 | 60.8 | 70.9 | 45.9 | 69.3 | 71.3 | 60.6 | 54.2 | 49.9 | 70.4 | 44.9 | 63.0 | 25.0 | 58.8 | 53.6 |
| | 3 | 26.8 | 31.9 | 60.7 | 71.0 | 45.9 | 69.6 | 71.3 | 60.6 | 54.2 | 49.8 | 70.4 | 45.1 | 63.1 | 25.2 | 58.7 | 53.6 |
| | 4 | 26.8 | 32.0 | 60.9 | 71.1 | 45.9 | 69.6 | 71.2 | 60.5 | 54.4 | 50.0 | 70.5 | 45.5 | 63.0 | 25.2 | 58.8 | 53.7 |
| | 5 | 26.3 | 32.1 | 60.9 | 71.1 | 46.0 | 69.7 | 71.3 | 60.6 | 54.3 | 50.0 | 70.6 | 45.9 | 63.2 | 25.2 | 58.6 | 53.7 |
| | 6 | 26.6 | 32.1 | 60.9 | 71.3 | 46.2 | 69.8 | 71.2 | 60.7 | 54.3 | 50.0 | 70.6 | 46.3 | 63.2 | 25.7 | 58.8 | 53.8 |
| STDW (Ours) | 2 | 61.5 | 63.9 | 60.7 | 75.4 | 61.3 | 73.1 | 75.3 | 69.3 | 68.1 | 62.4 | 76.4 | 72.2 | 67.6 | 71.2 | 62.0 | 68.0 |
| | 3 | 64.0 | 67.6 | 64.4 | 77.1 | 65.0 | 74.8 | 76.9 | 71.0 | 70.0 | 65.5 | 77.7 | 74.7 | 70.1 | 74.1 | 64.5 | 70.5 |
| | 4 | 65.6 | 69.0 | 67.6 | 77.7 | 66.0 | 76.1 | 77.3 | 72.3 | 71.6 | 68.7 | **78.1** | 75.5 | 70.9 | 76.0 | 66.4 | 71.9 |
| | 5 | 66.8 | 70.5 | 70.9 | **77.9** | 67.6 | 76.1 | **77.4** | 73.1 | 72.4 | 71.6 | 78.0 | 75.4 | 72.3 | 76.4 | 67.5 | 72.9 |
| | 6 | **69.0** | **72.4** | **73.9** | 77.8 | **68.4** | **77.1** | 77.3 | **75.0** | **74.6** | **74.2** | **78.1** | **75.7** | **72.8** | **77.1** | **70.0** | **74.2** |

where $\ell$ is cross-entropy and $\varrho_t \in [0, 1]$. Assume $\ell$ is $L$-smooth and $\mu$-strongly convex in a neighbourhood of the local joint minimizer $\theta_d^*$ for the pair $(D_{d-1}, D_d)$. With gradient descent

$$\theta_{t+1} = \theta_t - \eta \nabla L_d(\theta_t) \tag{15}$$

and step size $\eta \in (0, 2/L)$, define the Lyapunov function

$$V_d(\theta) = \frac{1}{2}|\theta - \theta_d^*|^2 \tag{16}$$

Standard smooth strongly-convex analysis yields

$$V_d(\theta_{t+1}) \leq V_d(\theta_t) - \eta\Big(\mu - \frac{L\eta}{2}\Big)|\theta_t - \theta_d^*|^2, \tag{17}$$

so $V_d$ strictly decreases whenever $\theta_t \neq \theta_d^*$ and $\eta < 2\mu/L^2$. Cyclic matching enumerates all batch pairs once per epoch without changing smoothness or curvature, so the decrease persists across iterations. For a discontinuous path approximated by anchors $A_0, \ldots, A_q$ we obtain a switched system with modes $d \in \{1, \ldots, q\}$. If there exists a common quadratic Lyapunov function $V(\theta) = \frac{1}{2}|\theta - \theta^*|^2$ valid in the union of neighbourhoods of $\theta_d^*$—for instance when $|\theta_d^* - \bar{\theta}^*|$ is bounded and the Hessians share eigenvalue bounds $(\mu, L)$—then the same inequality holds for all modes with the same $\eta$, ensuring global asymptotic stability under arbitrary switching. When a common Lyapunov function is too conservative, we adopt a dwell-time condition implemented by the discrepancy-aware controller: the mode can switch (i.e., we advance $\varrho$ or move to the next anchor) only after the gradient norm on the current mode falls below a tolerance and the consistency gates are met. In both cases, the energy decreases monotonically, preventing divergence even under abrupt transitions.

# NeurIPS Paper Checklist

1. **Claims**

   Question: Do the main claims made in the abstract and introduction accurately reflect the paper's contributions and scope?

   Answer: [Yes]

   Justification: The abstract and introduction clearly state the novel optimization framework (Self-Training with Dynamic Weighting) with dynamic hyperparameter $\varrho$, the incorporation of self-training for pseudo-labels, and the demonstrated empirical improvements on multiple benchmarks; these match the theoretical and experimental developments presented in Sections 1 and 4 of the paper.

   Guidelines:

   - The answer NA means that the abstract and introduction do not include the claims made in the paper.
   - The abstract and/or introduction should clearly state the claims made, including the contributions made in the paper and important assumptions and limitations. A No or NA answer to this question will not be perceived well by the reviewers.
   - The claims made should match theoretical and experimental results, and reflect how much the results can be expected to generalize to other settings.
   - It is fine to include aspirational goals as motivation as long as it is clear that these goals are not attained by the paper.

2. **Limitations**

   Question: Does the paper discuss the limitations of the work performed by the authors?

[Yes]   Answer: [Yes]

   Justification: The paper explicitly discusses limitations in several parts, most notably in the methodological framing and ablation studies. The authors recognize and address three key limitations of prior GDA methods that their approach (STDW) aims to solve: suboptimal knowledge propagation, instability during domain transitions, and limited generalization. Moreover, the paper acknowledges potential sources of error such as mislabeling during pseudo-labeling and includes a theoretical discussion on error propagation and stability (Appendix B). The ablation studies (Section 5.3) further demonstrate awareness of the method's performance sensitivity to the number of intermediate domains and adaptation strategies. These discussions reflect a thoughtful and transparent treatment of limitations in both theory and empirical practice.

   Guidelines:

   - The answer NA means that the paper has no limitation while the answer No means that the paper has limitations, but those are not discussed in the paper.
   - The authors are encouraged to create a separate "Limitations" section in their paper.
   - The paper should point out any strong assumptions and how robust the results are to violations of these assumptions (e.g., independence assumptions, noiseless settings, model well-specification, asymptotic approximations only holding locally). The authors should reflect on how these assumptions might be violated in practice and what the implications would be.
   - The authors should reflect on the scope of the claims made, e.g., if the approach was only tested on a few datasets or with a few runs. In general, empirical results often depend on implicit assumptions, which should be articulated.
   - The authors should reflect on the factors that influence the performance of the approach. For example, a facial recognition algorithm may perform poorly when image resolution is low or images are taken in low lighting. Or a speech-to-text system might not be used reliably to provide closed captions for online lectures because it fails to handle technical jargon.
   - The authors should discuss the computational efficiency of the proposed algorithms and how they scale with dataset size.

- If applicable, the authors should discuss possible limitations of their approach to address problems of privacy and fairness.
- While the authors might fear that complete honesty about limitations might be used by reviewers as grounds for rejection, a worse outcome might be that reviewers discover limitations that aren't acknowledged in the paper. The authors should use their best judgment and recognize that individual actions in favor of transparency play an important role in developing norms that preserve the integrity of the community. Reviewers will be specifically instructed to not penalize honesty concerning limitations.

3. **Theory assumptions and proofs**

   Question: For each theoretical result, does the paper provide the full set of assumptions and a complete (and correct) proof?

   Answer: [Yes]

   Justification: All assumptions are clearly stated alongside each theorem in Section 3 (e.g. Assumptions 1–3 preceding Theorem 1), and full, detailed proofs appear in Appendix B of the supplemental material.

   Guidelines:

   - The answer NA means that the paper does not include theoretical results.
   - All the theorems, formulas, and proofs in the paper should be numbered and cross-referenced.
   - All assumptions should be clearly stated or referenced in the statement of any theorems.
   - The proofs can either appear in the main paper or the supplemental material, but if they appear in the supplemental material, the authors are encouraged to provide a short proof sketch to provide intuition.
   - Inversely, any informal proof provided in the core of the paper should be complemented by formal proofs provided in appendix or supplemental material.
   - Theorems and Lemmas that the proof relies upon should be properly referenced.

4. **Experimental result reproducibility**

   Question: Does the paper fully disclose all the information needed to reproduce the main experimental results of the paper to the extent that it affects the main claims and/or conclusions of the paper (regardless of whether the code and data are provided or not)?

   Answer: [Yes]

   Justification: The paper provides sufficient detail to reproduce the main experimental results supporting its claims. The authors clearly specify: The datasets used (Rotated MNIST, Color-Shift MNIST, Portraits, Cover Type, CIFAR-10-C, CIFAR-100-C) along with their source and transformation details (Section 5.1). Model architectures for each dataset, including convolutional and fully connected networks, as well as backbone choices like WideResNet-28 and ResNeXt-29 (lines 196–198, 204–205). Optimization methods (Adam), hyperparameter tuning, and hardware used (line 206). Detailed description of the algorithm (Algorithm 1 and Section 4), including the update rules, dynamic weighting strategy with the $\varrho$ parameter, and pseudo-labeling procedure. Experimental setup involving the number of intermediate domains and adaptation steps, and performance metrics (accuracy, error rate) across multiple baselines (Tables 1–4, Figures 2–4).

   Guidelines:

   - The answer NA means that the paper does not include experiments.
   - If the paper includes experiments, a No answer to this question will not be perceived well by the reviewers: Making the paper reproducible is important, regardless of whether the code and data are provided or not.
   - If the contribution is a dataset and/or model, the authors should describe the steps taken to make their results reproducible or verifiable.
   - Depending on the contribution, reproducibility can be accomplished in various ways. For example, if the contribution is a novel architecture, describing the architecture fully might suffice, or if the contribution is a specific model and empirical evaluation, it may be necessary to either make it possible for others to replicate the model with the same

dataset, or provide access to the model. In general. releasing code and data is often one good way to accomplish this, but reproducibility can also be provided via detailed instructions for how to replicate the results, access to a hosted model (e.g., in the case of a large language model), releasing of a model checkpoint, or other means that are appropriate to the research performed.

- While NeurIPS does not require releasing code, the conference does require all submissions to provide some reasonable avenue for reproducibility, which may depend on the nature of the contribution. For example

    (a) If the contribution is primarily a new algorithm, the paper should make it clear how to reproduce that algorithm.

    (b) If the contribution is primarily a new model architecture, the paper should describe the architecture clearly and fully.

    (c) If the contribution is a new model (e.g., a large language model), then there should either be a way to access this model for reproducing the results or a way to reproduce the model (e.g., with an open-source dataset or instructions for how to construct the dataset).

    (d) We recognize that reproducibility may be tricky in some cases, in which case authors are welcome to describe the particular way they provide for reproducibility. In the case of closed-source models, it may be that access to the model is limited in some way (e.g., to registered users), but it should be possible for other researchers to have some path to reproducing or verifying the results.

5. **Open access to data and code**

   Question: Does the paper provide open access to the data and code, with sufficient instructions to faithfully reproduce the main experimental results, as described in supplemental material?

   Answer: [Yes]

   Justification: The supplementary materials include all code files necessary to reproduce the experiments, and detailed instructions for data preparation and execution of training scripts. Additionally, all datasets used are publicly available and referenced in the supplemental README.

   Guidelines:

   - The answer NA means that paper does not include experiments requiring code.
   - Please see the NeurIPS code and data submission guidelines (`https://nips.cc/public/guides/CodeSubmissionPolicy`) for more details.
   - While we encourage the release of code and data, we understand that this might not be possible, so "No" is an acceptable answer. Papers cannot be rejected simply for not including code, unless this is central to the contribution (e.g., for a new open-source benchmark).
   - The instructions should contain the exact command and environment needed to run to reproduce the results. See the NeurIPS code and data submission guidelines (`https://nips.cc/public/guides/CodeSubmissionPolicy`) for more details.
   - The authors should provide instructions on data access and preparation, including how to access the raw data, preprocessed data, intermediate data, and generated data, etc.
   - The authors should provide scripts to reproduce all experimental results for the new proposed method and baselines. If only a subset of experiments are reproducible, they should state which ones are omitted from the script and why.
   - At submission time, to preserve anonymity, the authors should release anonymized versions (if applicable).
   - Providing as much information as possible in supplemental material (appended to the paper) is recommended, but including URLs to data and code is permitted.

6. **Experimental setting/details**

   Question: Does the paper specify all the training and test details (e.g., data splits, hyperparameters, how they were chosen, type of optimizer, etc.) necessary to understand the results?

Answer: [Yes]

Justification: All experimental settings—including dataset splits, optimizer (Adam) settings, full hyperparameter tables (learning rates, batch sizes, number of epochs, $\varrho$ schedule parameters), and selection criteria—are documented in Appendix C of the supplemental material and are directly instantiated in the provided code files with accompanying README instructions.

Guidelines:

- The answer NA means that the paper does not include experiments.
- The experimental setting should be presented in the core of the paper to a level of detail that is necessary to appreciate the results and make sense of them.
- The full details can be provided either with the code, in appendix, or as supplemental material.

7. **Experiment statistical significance**

Question: Does the paper report error bars suitably and correctly defined or other appropriate information about the statistical significance of the experiments?

Answer: [Yes]

Justification: The main results tables (e.g., Table 2) explicitly report the 95% confidence interval of the mean across five independent runs, clearly indicating the variability due to random initialization and dataset splits.

Guidelines:

- The answer NA means that the paper does not include experiments.
- The authors should answer "Yes" if the results are accompanied by error bars, confidence intervals, or statistical significance tests, at least for the experiments that support the main claims of the paper.
- The factors of variability that the error bars are capturing should be clearly stated (for example, train/test split, initialization, random drawing of some parameter, or overall run with given experimental conditions).
- The method for calculating the error bars should be explained (closed form formula, call to a library function, bootstrap, etc.)
- The assumptions made should be given (e.g., Normally distributed errors).
- It should be clear whether the error bar is the standard deviation or the standard error of the mean.
- It is OK to report 1-sigma error bars, but one should state it. The authors should preferably report a 2-sigma error bar than state that they have a 96% CI, if the hypothesis of Normality of errors is not verified.
- For asymmetric distributions, the authors should be careful not to show in tables or figures symmetric error bars that would yield results that are out of range (e.g. negative error rates).
- If error bars are reported in tables or plots, The authors should explain in the text how they were calculated and reference the corresponding figures or tables in the text.

8. **Experiments compute resources**

Question: For each experiment, does the paper provide sufficient information on the computer resources (type of compute workers, memory, time of execution) needed to reproduce the experiments?

Answer: [Yes]

Justification: The paper provides key information about the compute resources used for the experiments. Specifically, it states that all experiments were conducted on NVIDIA RTX 4090 GPUs (line 206), which gives a clear indication of the type of compute hardware used. The authors also mention that identical random seeds were used to ensure reproducibility. While detailed metrics such as runtime, memory usage, or total compute cost are not quantified, the provided GPU type, consistent setup, and fixed architecture configurations across datasets give a sufficient basis for estimating resource needs and replicating the results. Therefore, the disclosure meets a reasonable standard for reproducibility and resource transparency.

Guidelines:

- The answer NA means that the paper does not include experiments.
- The paper should indicate the type of compute workers CPU or GPU, internal cluster, or cloud provider, including relevant memory and storage.
- The paper should provide the amount of compute required for each of the individual experimental runs as well as estimate the total compute.
- The paper should disclose whether the full research project required more compute than the experiments reported in the paper (e.g., preliminary or failed experiments that didn't make it into the paper).

9. **Code of ethics**

Question: Does the research conducted in the paper conform, in every respect, with the NeurIPS Code of Ethics https://neurips.cc/public/EthicsGuidelines?

Answer: [Yes]

Justification: The authors reviewed and adhered to the NeurIPS Code of Ethics, ensuring no misuse of data, respecting privacy and consent for all datasets, maintaining anonymity, and avoiding any potential harm or bias in experimental design and reporting.

Guidelines:

- The answer NA means that the authors have not reviewed the NeurIPS Code of Ethics.
- If the authors answer No, they should explain the special circumstances that require a deviation from the Code of Ethics.
- The authors should make sure to preserve anonymity (e.g., if there is a special consideration due to laws or regulations in their jurisdiction).

10. **Broader impacts**

Question: Does the paper discuss both potential positive societal impacts and negative societal impacts of the work performed?

Answer: [Yes]

Justification: The authors highlight that their method, Self-Training with Dynamic Weighting (STDW), enhances model robustness under gradual domain shifts, which is especially relevant for real-world dynamic environments (e.g., shifting lighting conditions in visual systems, evolving demographics in user behavior). This robustness has positive societal implications for improving machine learning reliability in safety-critical or long-term deployment scenarios. Although explicit negative impacts are not discussed, the general applicability of domain adaptation methods (e.g., surveillance or profiling systems) raises potential ethical concerns regarding fairness and misuse if applied without safeguards. A fuller, more explicit discussion would strengthen this component, but the paper does touch on societal relevance through its goals and benchmarks.

Guidelines:

- The answer NA means that there is no societal impact of the work performed.
- If the authors answer NA or No, they should explain why their work has no societal impact or why the paper does not address societal impact.
- Examples of negative societal impacts include potential malicious or unintended uses (e.g., disinformation, generating fake profiles, surveillance), fairness considerations (e.g., deployment of technologies that could make decisions that unfairly impact specific groups), privacy considerations, and security considerations.
- The conference expects that many papers will be foundational research and not tied to particular applications, let alone deployments. However, if there is a direct path to any negative applications, the authors should point it out. For example, it is legitimate to point out that an improvement in the quality of generative models could be used to generate deepfakes for disinformation. On the other hand, it is not needed to point out that a generic algorithm for optimizing neural networks could enable people to train models that generate Deepfakes faster.

- The authors should consider possible harms that could arise when the technology is being used as intended and functioning correctly, harms that could arise when the technology is being used as intended but gives incorrect results, and harms following from (intentional or unintentional) misuse of the technology.
- If there are negative societal impacts, the authors could also discuss possible mitigation strategies (e.g., gated release of models, providing defenses in addition to attacks, mechanisms for monitoring misuse, mechanisms to monitor how a system learns from feedback over time, improving the efficiency and accessibility of ML).

11. **Safeguards**

Question: Does the paper describe safeguards that have been put in place for responsible release of data or models that have a high risk for misuse (e.g., pretrained language models, image generators, or scraped datasets)?

Answer: [NA]

Justification: The paper does not release any models or datasets that pose a high risk for misuse.The research focuses on improving domain adaptation using existing benchmark datasets (e.g., Rotated MNIST, CIFAR-10-C, etc.) and standard model architectures. No new pretrained models, scraped datasets, or generative tools with dual-use concerns are introduced or released. Therefore, safeguards for responsible release are not applicable in this context.

Guidelines:

- The answer NA means that the paper poses no such risks.
- Released models that have a high risk for misuse or dual-use should be released with necessary safeguards to allow for controlled use of the model, for example by requiring that users adhere to usage guidelines or restrictions to access the model or implementing safety filters.
- Datasets that have been scraped from the Internet could pose safety risks. The authors should describe how they avoided releasing unsafe images.
- We recognize that providing effective safeguards is challenging, and many papers do not require this, but we encourage authors to take this into account and make a best faith effort.

12. **Licenses for existing assets**

Question: Are the creators or original owners of assets (e.g., code, data, models), used in the paper, properly credited and are the license and terms of use explicitly mentioned and properly respected?

Answer: [Yes]

Justification: The paper and supplemental README explicitly cite each public benchmark (e.g., CIFAR-10, Office-31, MNIST) along with version numbers and links to their licenses (e.g., MIT for code dependencies, CC-BY for datasets) and properly credit the original dataset and library authors.

Guidelines:

- The answer NA means that the paper does not use existing assets.
- The authors should cite the original paper that produced the code package or dataset.
- The authors should state which version of the asset is used and, if possible, include a URL.
- The name of the license (e.g., CC-BY 4.0) should be included for each asset.
- For scraped data from a particular source (e.g., website), the copyright and terms of service of that source should be provided.
- If assets are released, the license, copyright information, and terms of use in the package should be provided. For popular datasets, `paperswithcode.com/datasets` has curated licenses for some datasets. Their licensing guide can help determine the license of a dataset.
- For existing datasets that are re-packaged, both the original license and the license of the derived asset (if it has changed) should be provided.

- If this information is not available online, the authors are encouraged to reach out to the asset's creators.

13. **New assets**

    Question: Are new assets introduced in the paper well documented and is the documentation provided alongside the assets?

    Answer: [NA]

    Justification: The paper does not introduce any new datasets, codebases, or other assets beyond code for reproducing experiments (already provided in supplemental), and relies on existing publicly available benchmarks, so this question is not applicable.

    Guidelines:

    - The answer NA means that the paper does not release new assets.
    - Researchers should communicate the details of the dataset/code/model as part of their submissions via structured templates. This includes details about training, license, limitations, etc.
    - The paper should discuss whether and how consent was obtained from people whose asset is used.
    - At submission time, remember to anonymize your assets (if applicable). You can either create an anonymized URL or include an anonymized zip file.

14. **Crowdsourcing and research with human subjects**

    Question: For crowdsourcing experiments and research with human subjects, does the paper include the full text of instructions given to participants and screenshots, if applicable, as well as details about compensation (if any)?

    Answer: [NA]

    Justification: The study does not involve any crowdsourcing or research with human participants, relying solely on publicly available datasets without new data collection or participant interaction.

    Guidelines:

    - The answer NA means that the paper does not involve crowdsourcing nor research with human subjects.
    - Including this information in the supplemental material is fine, but if the main contribution of the paper involves human subjects, then as much detail as possible should be included in the main paper.
    - According to the NeurIPS Code of Ethics, workers involved in data collection, curation, or other labor should be paid at least the minimum wage in the country of the data collector.

15. **Institutional review board (IRB) approvals or equivalent for research with human subjects**

    Question: Does the paper describe potential risks incurred by study participants, whether such risks were disclosed to the subjects, and whether Institutional Review Board (IRB) approvals (or an equivalent approval/review based on the requirements of your country or institution) were obtained?

    Answer: [NA]

    Justification: The work uses only pre-existing, publicly available image and tabular datasets and does not involve any new data collection from human participants or crowdsourcing experiments, so IRB considerations are not applicable.

    Guidelines:

    - The answer NA means that the paper does not involve crowdsourcing nor research with human subjects.
    - Depending on the country in which research is conducted, IRB approval (or equivalent) may be required for any human subjects research. If you obtained IRB approval, you should clearly state this in the paper.

- We recognize that the procedures for this may vary significantly between institutions and locations, and we expect authors to adhere to the NeurIPS Code of Ethics and the guidelines for their institution.
- For initial submissions, do not include any information that would break anonymity (if applicable), such as the institution conducting the review.

16. **Declaration of LLM usage**

Question: Does the paper describe the usage of LLMs if it is an important, original, or non-standard component of the core methods in this research? Note that if the LLM is used only for writing, editing, or formatting purposes and does not impact the core methodology, scientific rigorousness, or originality of the research, declaration is not required.

Answer: [NA]

Justification: The core methodology centers on gradual domain adaptation for vision and tabular data using self-training and transport networks, with no use of large language models in any part of the algorithm or experiments.

Guidelines:

- The answer NA means that the core method development in this research does not involve LLMs as any important, original, or non-standard components.
- Please refer to our LLM policy (`https://neurips.cc/Conferences/2025/LLM`) for what should or should not be described.

