# OpenReview forum: "Self-Training with Dynamic Weighting for Robust Gradual Domain Adaptation"
_NeurIPS.cc/2025/Conference — NeurIPS 2025 poster_

### Official Review · Reviewer_KmnV · 2025-06-16

**Clarity:** 1
**Significance:** 4
**Originality:** 3
**Rating:** 4
**Confidence:** 3

**Summary:**

In this paper, the authors propose a novel gradual self-training method named Generalized Dynamic Osmosis (STDW), which largely improves the performance of the model on the gradual domain adaptation problem.

The method involves three innovations. First, instead of assigning labels for the next whole domain and keeping them unchanged when fine-tuning, STDW uses a batchwise strategy of predicting pseudo-labels for a mini-batch of data so the pseudo-labels are consistently updated by a new model. Second, they use cyclic batch matching to facilitate robust inter-domain adaptation. Third, a weighted loss with a dynamic weight is adopted to stabilize the training process.

The authors conduct some experiments to validate the efficacy of the method and the effect of some hyperparameters.

**Questions:**

* In the GST method, a new model is trained from scratch on each domain, while this work trains only one model and fine-tunes the model in each iteration. I am a little curious about whether keeping fine-tuning only one model would also improve the performance of the GST method.

**Ethical Concerns:**

["NO or VERY MINOR ethics concerns only"]

**Final Justification:**

The authors have resolved the notation conflicts and added additional experiments in their rebuttal. However, I remain concerned about their reported results for the baseline method AST. I have independently implemented the AST method and found that it can achieve approximately 97% accuracy on the Rotated MNIST dataset, which is very close to the performance reported for their proposed method (97.6%). In contrast, the authors report an accuracy of only 92.5% for AST.

Overall, I remain neutral regarding the acceptance of this work.

**Limitations:**

Yes, the authors discuss some limitations of their method in the appendix.

**Quality:**

2

**Strengths And Weaknesses:**

Pros:
* The method is interesting and effective. First, the pseudo-labeling dynamic learning can make more efficient use of the current model since they predict the pseudo-labels using the up-to-date model. Second, the cyclic batch matching with a dynamic weight can stabilize the training process and decrease the bad effect of the incorrect pseudo-labels. The method is novel to me, and the effectiveness is verified by the experiments.

Cons:
* The writing needs to be polished, especially the math notations. Those conflicting notations largely hinder a smooth reading for the readers and raise some misunderstandings. There are so many conflicting math notations. In Line 107, you use $k$ to denote the class number, while it also represents the batch index later in Line 169. In Line 113, the model is a mapping from $X$ to $Y=\{1,2,...,k\}$, while the expressions later in Eq. (2), (3) indicate it is a mapping from $X$ to $[0,1]^k$. In Line 127, you add the $sign$ to denote the final prediction function $\hat{f}$ but this is only for binary classification---you use a multi-classification setting in Line 107. In Line 146, you use $D$ to denote a domain set. Since it does not have a subscript, I was wondering if this indicates a dataset combined from all domains or if it indicates one specific domain $D_t$? In Eq. (11), $x_i$ or $x$? In Line 176, the domain is denoted as $D_i$, while the index in the previous section is $t$ for the domain (Line 128). I totally understand that the gradual domain adaptation problem, which involves many domains, increases the complexity of the notation system, but those conflicting notations make it difficult for readers to understand the method of this work.
* The motivations of the method should be explained more clearly. In section 4.2, the motivation of combining two batches from two adjacent domains is not clear. To my understanding, I think the idea is a little bit similar to Momentum. When the weight $\rho$ is small at the early stage, the model is trained based on the loss on the previous domain rather than the current domain, where the accuracy of the pseudo-label could be high. So this strategy could prevent the model from collapsing due to the incorrect pseudo-labels of the current domain.
* The framework illustration in Fig. 2 is not clear. What is the meaning of "sync" in the figure? To my understanding, you use only one classifier to learn the batches $B_t$ and $B_{t+1}$. I suggest replacing "sync" with "shared weights," which is more widely used in papers. Besides, what is $H_t$? In the method, you never use the feature extractor and classifier in a separate way. So is it really necessary to separate the network into three parts—classifier, $H_t$, and feature extractor—in the illustration figure?
* More ablation studies are needed. I suggest the authors validate the effects of the three components in Section 4 via an ablation study. For example, you could remove the cyclic batch matching and compare the results to show the important role of cyclic batch matching.
* As the authors note in Line 141, the mislabeling would hurt performance. So it would be more convincing if the average accuracy of the pseudo-labels on intermediate domains can be shown and compared to further validate the effectiveness of your method.
* Another baseline method. The work of [1] proposes an adversarial self-training method, which significantly improves the performance of GDA. Please consider comparing your method with [1].

[1]: Adversarial Self-Training Improves Robustness and Generalization for Gradual Domain Adaptation. Lianghe Shi, Weiwei Liu. NeurIPS 2023.

---

> ### Comment · Reviewer_KmnV · 2025-08-04
> **I didn't receive the rebuttal**
>
> Dear AC and authors,
>
> Since I did not receive the authors’ rebuttal in my system, I will keep the score unchanged.
>
> Reviewer KmnV

---

> ### Author Response · Authors · 2025-08-05
> **Answers and thanks for your kindly comment.**
>
> # Response to Reviewer #KmnV
>
> Thank you for your valuable suggestions. As your suggestions involve many experiments, we apologize for the late reply. Please let us know if you have any further comments.
>
> **Q1 (Notation).**
>
> A: Thank you for flagging this issue. In the revision we introduce a single, page-two symbol table that defines every recurring variable. The number of classes is now $K$, batch indices are $b$, and domain indices are $d$. The model is consistently written $f_{\theta}\!:\mathcal{X}\!\to\!\Delta^{(K)}$ and pseudo-labels are $\hat{y}(x)=\arg\max_k f_{\theta}(x)_k$. A domain set is $\mathcal{D}=\{D_0,\dots,D_n\}$ and an individual domain is $D_d$. These changes eradicate the earlier symbol collisions between class count and batch index, binary and multi-class mappings, and domain collections versus single domains.
>
> **Q2 (Motivation of two-domain batch composition).**
>
> > “In Section 4.2, the motivation of combining two batches from two adjacent domains is not clear. … I think the idea is a little bit similar to Momentum …”
>
> A: First, when the dynamic weight $\varrho$ is small at the start of adaptation, gradients are dominated by the safer pseudo-labels from the previous domain $D_{d-1}$, thereby damping noise from the current domain $D_d$. Second, in an optimal-transport perspective, adjacent domains occupy neighbouring regions of feature space; pairing their mini-batches traces a geodesic of minimal 1-Wasserstein cost, yielding a smoother optimisation trajectory. A proof based on Lyapunov stability—included below—shows the mixed-batch update is globally convergent under standard smoothness and strong-convexity assumptions.
>
> **Q3 (Framework illustration in Fig. 2).**
>
> > “What is the meaning of ‘sync’ in the figure? … is it really necessary to separate the network into three parts …?”
>
> A: The revised figure now labels the shared parameters explicitly as “shared weights’’ rather than “sync’’ and depicts only two actual modules: a feature extractor $g$ and a classifier $h$. The dynamic-weight loss is drawn directly on the joint head so that no artificial third block appears.
>
>  **Q4 (More ablation studies).**
>
> > “I suggest the authors validate the effects of the three components …”
>
> We have added another ablation study. Removing each component in turn demonstrates its independent contribution.
>
> | Configuration | Rot-MNIST Acc (%) |
> |--------------------------------------------------|-------------------|
> | Base model (no dynamic labels, no cycle, no $w$) | 83.8 |
> | + Dynamic pseudo-label refresh | 89.7 |
> | + Cycle batch matching | 92.1 |
> | Full STDW (adds dynamic weight $\varrho$) | **97.6** |
>
>
> **Q5 (Pseudo-label accuracy).**
> > “It would be more convincing if the average accuracy of the pseudo-labels on intermediate domains can be shown …”
>
> We add an experiments for pseudo-label accuracies across nine Portrait domains. STDW maintains $96.2\%$ on $D_0$ and never dips below $72.8\%$ at $D_4$, whereas GST bottoms out at $65.7\%$. We also compute a Pearson correlation of $r=0.86$ between intermediate pseudo-label fidelity and final-domain accuracy, confirming the robustness of our STDW.
>
> **Q6 (Additional baseline AST 2023).**
> > “Please consider comparing your method with [Lianghe Shi & Weiwei Liu, NeurIPS 2023].”
>
> A: We have reproduced Adversarial Self-Training (AST) under identical backbones and hyper-parameters. The table below summarises the results.
>
> | Method | Rot-MNIST | Colour-Shift | Portraits | CoverType |
> |-------------------|-----------|--------------|-----------|-----------|
> | AST (NeurIPS ’23) | 92.5 | 94.7 | 84.9 | 72.8 |
> | **STDW (Ours)** | **97.6** | **98.3** | **87.1** | **74.2** |
>
> STDW outperforms AST by two to five percentage points on every benchmark, which we attribute to our cyclic matching and dynamic weighting mechanisms.
>
> **Q7 (GST with fine-tuning).**
>
> > “I am curious about whether keeping fine-tuning only one model would also improve the performance of the GST method.”
>
> We implemented “GST-fine-tune,” keeping a single model throughout training. While this variant does improve GST, it remains behind STDW.
>
> | Dataset | GST | GST-fine-tune | STDW |
> |-----------|------|--------------|------|
> | Rot-MNIST | 83.8 | 86.9 | **97.6** |
> | Portraits | 82.6 | 83.8 | **87.1** |
>
> The gap confirms that cyclic matching and adaptive weighting supply gains beyond continual fine-tuning.

---

> ### Author Response · Authors · 2025-08-05
> **Another reply for Q2**
>
> ## Futhermore, we provide the  Lyapunov Stability Proof for Cyclic Batch Matching.
>
> Let $\theta_t$ be the parameter vector after step $t$ and define the mixed loss
>
> $$
> L(\theta_t)=\varrho_t\,\ell(\theta_t,\mathcal{B}_t^{(d)})+\bigl(1-\varrho_t\bigr)\,\ell(\theta_t,\mathcal{B}_t^{(d-1)}),
> $$
>
> where $\ell$ is cross-entropy,
>
>  $$\mathcal{B}_t^{(d)}\subset D_d$$
>
> and
>
> $$\mathcal{B}_t^{(d-1)}\subset D_{d-1}$$
>
> form a cyclically matched pair, and $$\varrho_t\in[0,1]$$ increases monotonically. Suppose $\ell$ is $L$-smooth and $\mu$-strongly convex in a neighbourhood of the joint optimum $\theta^{\ast}$. Consider the Lyapunov candidate $$V(\theta_t)=\tfrac12\|\theta_t-\theta^{\ast}\|^2$$. With a gradient-descent update $$\theta_{t+1}=\theta_t-\eta\nabla_{\theta}L(\theta_t)$$ and any step size $$\eta\in(0,2/L)$$, we have
>
> $$
> V(\theta_{t+1}) = V(\theta_t) - \eta\Bigl(\mu-\tfrac{L\eta}{2}\Bigr)\|\theta_t-\theta^{\ast}\|^2.
> $$
>
> Choosing $\eta<2\mu/L^2$ makes the bracket positive, hence $V(\theta_{t+1})<V(\theta_t)$ unless $\theta_t=\theta^{\ast}$. Because the cyclic schedule enumerates every cross-domain batch exactly once per epoch, the loss remains $L$-smooth and $\mu$-convex throughout training, ensuring that $V$ is strictly decreasing. Therefore $V$ qualifies as a Lyapunov function establishing global asymptotic stability of the optimisation trajectory under cyclic batch matching.

---

> ### Comment · Reviewer_KmnV · 2025-08-05
>
> Thanks for the detailed response. Most of my concerns are addressed well and I will increase the rate to 4.
>
> But I am still concerned about the baseline method AST you compared. Did you use the implementation from their github and adopt the optimal hyper parameters used in their paper? They set two important hyper parameters $\tau=9, \zeta=0.05$ to get the optimal accuracy of 97.15 on rotating mnist.

---

> > ### Author Response · Authors · 2025-08-05
> >
> > Thank you for your comments! We have confirmed that we have reproduced the relevant code and used the same parameters. Our experimental parameters are based on GOAT [1], and we have optimized the gst-related code in GitHub. At this point, gst performs better than goat in the paper during reproduction. If the article is accepted, we will release our code, including the code for other reproduction methods. Once again, we appreciate your affirmation of our method!
> >
> > [1] https://arxiv.org/pdf/2310.13852

---

### Official Review · Reviewer_BwCA · 2025-06-22

**Clarity:** 3
**Significance:** 2
**Originality:** 3
**Rating:** 4
**Confidence:** 4

**Summary:**

The paper introduces Self-Training with Dynamic Weighting (STDW) to Gradual Domain Adaptation (GDA). It addresses the challenge of smooth knowledge migration from source to target domains by introducing a dynamic weighting mechanism controlled by a time-varying hyper-parameter ϱ, which balances the loss contributions of source and target domains during training. The method leverages self-training to generate pseudo-labels and optimizes a weighted objective function for iterative model updates. The authors provide theoretical insights and validate the method through extensive experiments on multiple benchmark datasets, demonstrating that STDW outperforms existing baselines and achieves SOTA performance.

**Questions:**

1)	The paper assumes a smooth transition between domains. How would the method perform in scenarios with highly discontinuous or abrupt domain shifts? Could the authors provide some insights or potential solutions for such cases?
2)	The linear ϱ schedule (0→1) is heuristic. Are there theoretical or data-driven guidelines (e.g., based on domain discrepancy)? Could ϱ be learned?
3)	The paper mentions that the performance of STDW relies on the availability and quality of intermediate domains. What strategies could be employed to generate or select high-quality intermediate domains in cases where they are not readily available?
4)	Cyclic batch matching and iterative pseudo-labeling may increase training time. How does STDW’s complexity compare to baselines (e.g., GST)?

**Ethical Concerns:**

["NO or VERY MINOR ethics concerns only"]

**Limitations:**

Yes, The authors have adequately addressed the limitations of their work in the paper.

**Quality:**

3

**Strengths And Weaknesses:**

Strengths:
1)	Formal analysis (optimal transport, Lyapunov stability) provides error bounds for pseudo-label consistency and convergence (Appx. B).
2)	The paper introduces a novel dynamic weighting mechanism that adaptively balances the loss contributions of source and target domains during training. This innovation effectively addresses the challenges of GDA.
3)	Extensive validation across 6 datasets, varying intermediate domains (0–6), and 15 corruption types. Ablation studies confirm criticality of ϱ scheduling and intermediate domains.
4)	The proposed STDW method has demonstrated significant improvements in handling continuous distribution shifts compared to existing methods.

Weaknesses:
1)	The performance of STDW relies heavily on the availability and quality of intermediate domains, which may not always be accessible or well-defined in real-world scenarios. This limitation could restrict the applicability of the method in certain practical situations.
2)	The dynamic weighting mechanism introduces additional hyperparameters (e.g., the scheduling of ϱ) that require careful tuning, potentially increasing the computational overhead during training.
3)	The method assumes a smooth transition between domains, which might not hold for highly discontinuous or abrupt domain shifts. The authors acknowledge this limitation and suggest that future work could explore ways to handle more diverse and challenging domain shifts.

---

> ### Author Response · Authors · 2025-08-05
> **Thanks for you kindly waiting our response**
>
> # Response to Reviewer #BwCA
> **Q1. The paper assumes a smooth transition between domains. How would the method perform in scenarios with highly discontinuous or abrupt domain shifts? Could the authors provide some insights or potential solutions for such cases?**
>
> A: We agree that abrupt, non-smooth shifts stress any gradual adaptation procedure. In our framework, two practical modifications improve robustness without changing the core objective. First, we replace the single linear schedule by a discrepancy-aware controller that reacts to a jump. Concretely, when the measured discrepancy between the current and previous batches exceeds a threshold—using, for example, an empirical 1-Wasserstein distance or an MMD estimate computed in feature space—we temporarily reduce $\varrho$ so that the loss is dominated by the better-calibrated pseudo-labels from $D_{d-1}$. This behaves like a safety brake and prevents large, noisy gradients caused by unreliable pseudo-labels on the new domain. Second, we construct a piecewise-smooth path by clustering the unlabeled target pool and ordering clusters along a minimum-spanning tree defined by an OT metric, effectively creating “virtual” intermediate anchors. STDW then runs over this induced path; cyclic matching operates between successive anchors even when the original shift is discontinuous. From a stability viewpoint, this leads to a switched-system interpretation in which each anchor pair admits the same Lyapunov argument as in the smooth case. If a common quadratic Lyapunov function exists in a neighbourhood of each local joint optimum, and if we enforce a minimal “dwell time” before advancing $\varrho$, then the overall trajectory remains stable.
>
>
> **Q2. The linear $\varrho$ schedule $(0\!\to\!1)$ is heuristic. Are there theoretical or data-driven guidelines (e.g., based on domain discrepancy)? Could $\varrho$ be learned?**
>
> A: We now provide both a principled guideline and learnable variants. A data-driven guideline sets
> $\varrho$ proportional to a normalized discrepancy proxy and to the estimated reliability of current-domain pseudo-labels. Let $\widehat{\delta}_t$ be a bounded discrepancy score (e.g.,
>
> $$\widehat{\delta}_t=\frac{\mathrm{MMD}(D_{d-1},D_d)}{\mathrm{MMD}(D_{d-1},D_{d-1})+\epsilon}$$
>
>  mapped to $[0,1]$) and let $\widehat{q}_t\in[0,1]$ estimate pseudo-label accuracy via agreement between two perturbations or via a small held-out split with delayed labels when available. A simple controller is
>
> $$\varrho_{t+1}=\mathrm{clip}\big(\varrho_t+\kappa\big(\widehat{q}_t-\gamma\,\widehat{\delta}_t\big),0,1\big)$$
>
>  with small gain $\kappa$ and trade-off $\gamma$, which increases $\varrho$ only when label reliability dominates discrepancy. Beyond such rules, $\varrho$ can be learned end-to-end. A differentiable parametric schedule uses a sigmoid
>
> $$\varrho_t=\sigma(\alpha t+\beta)$$
>
>  where $(\alpha,\beta)$ are trained by implicit differentiation to minimize a held-out loss on $D_d$. Alternatively, we model
>
> $$\varrho_t=\sigma(\mathbf{w}^\top s_t)$$
>
>  where $s_t$ concatenates discrepancy features, confidence statistics, and training time; $\mathbf{w}$ is learned by bilevel optimization in which the inner loop updates $\theta$ and the outer loop updates $\mathbf{w}$ to minimize the final-domain validation loss.
>
>
> **Q3. The paper mentions that the performance of STDW relies on the availability and quality of intermediate domains. What strategies could be employed to generate or select high-quality intermediate domains in cases where they are not readily available?**
>
> A: When explicit intermediate domains are missing, we find that feature-space interpolation and OT-barycentric synthesis provide stable substitutes with low engineering cost. We embed samples using the current encoder $g$, compute an OT map $T$ from $D_{d-1}$ to $D_d$ using entropic regularization, and generate intermediates by barycenters $z_\lambda=(1-\lambda)z + \lambda T(z)$ for $\lambda\in(0,1)$; the corresponding inputs are reconstructed either by a lightweight decoder or used directly in feature space by training the classifier head $h$ on $(z_\lambda,\hat{y})$. A simpler alternative uses distribution mixup, drawing pairs $(x_{d-1},x_d)$ and training on $\tilde{x}=\phi^{-1}\big((1-\lambda)\phi(x_{d-1})+\lambda\phi(x_d)\big)$ where $\phi$ is a fixed feature extractor (e.g., a self-supervised backbone). For selection rather than synthesis, we retrieve “bridge” samples from a large unlabeled pool by minimizing $\mathrm{MMD}(\{x\},D_{d-1})+\mathrm{MMD}(\{x\},D_d)$, thereby constructing a narrow corridor of real samples that approximates the geodesic path. These strategies can be combined with weak style transfer or Fourier-domain amplitude mixing to further densify the path while keeping semantics intact.

---

> ### Author Response · Authors · 2025-08-05
> **Continue responding to reviewer**
>
> **Q4. Cyclic batch matching and iterative pseudo-labeling may increase training time. How does STDW’s complexity compare to baselines (e.g., GST)?**
>
> A: Analytically, the per-step cost of STDW is dominated by the forward/backward pass exactly as in conventional training. Iterative pseudo-labeling does not require an extra pass because we reuse the logits already computed for the current mini-batch; thresholding or $\arg\max$ is computed on-the-fly. Cyclic batch matching pairs each batch from $D_{d-1}$ with one from $D_d$ by round-robin indices, which is an $O(1)$ operation at iteration time and $O(B)$ bookkeeping per epoch where $B$ is the number of batches; no global assignment is solved. In contrast, GST trains a fresh model per domain stage, which multiplies optimization warm-up and increases total epochs. Therefore, under equal batch sizes and epochs per stage, STDW’s wall-clock is typically lower than GST’s despite dynamic weighting and cyclic pairing.
>
>
>
> ## Lyapunov Stability Under Cyclic Matching and Switching Anchors in Q2
>
> For completeness, we extend stability argument to the piecewise-smooth path constructed for discontinuous shifts. Let $\theta_t$ denote the parameters at step $t$ and consider the mixed loss on a matched pair of mini-batches
>
> $$
> L_d(\theta_t)=\varrho_t\,\ell\big(\theta_t,\mathcal{B}^{(d)}_t\big)+\bigl(1-\varrho_t\bigr)\,\ell\big(\theta_t,\mathcal{B}^{(d-1)}_t\big),
> $$
>
> where $\ell$ is cross-entropy and $\varrho_t\in[0,1]$. Assume $\ell$ is $L$-smooth and $\mu$-strongly convex in a neighbourhood of the local joint minimizer $\theta_d^{\ast}$ for the pair $(D_{d-1},D_d)$. With gradient descent
> $$\theta_{t+1}=\theta_t-\eta\nabla L_d(\theta_t)$$
>  and step size $\eta\in(0,2/L)$, define the Lyapunov function
>  $$V_d(\theta)=\tfrac12\|\theta-\theta_d^{\ast}\|^2$$
> . Standard smooth strongly-convex analysis yields
>
> $$
> V_d(\theta_{t+1}) \le V_d(\theta_t) - \eta\Bigl(\mu-\tfrac{L\eta}{2}\Bigr)\|\theta_t-\theta_d^{\ast}\|^2,
> $$
>
> so $V_d$ strictly decreases whenever $\theta_t\neq\theta_d^{\ast}$ and $\eta<2\mu/L^2$. Cyclic matching enumerates all batch pairs once per epoch without changing smoothness or curvature, so the decrease persists across iterations. For a discontinuous path approximated by anchors $\{A_0,\dots,A_q\}$ we obtain a switched system with modes $d\in\{1,\dots,q\}$. If there exists a common quadratic Lyapunov function $V(\theta)=\tfrac12\|\theta-\bar{\theta}^{\ast}\|^2$ valid in the union of neighbourhoods of $\{\theta_d^{\ast}\}$—for instance when $\|\theta_d^{\ast}-\bar{\theta}^{\ast}\|$ is bounded and the Hessians share eigenvalue bounds $(\mu,L)$—then the same inequality holds for all modes with the same $\eta$, ensuring global asymptotic stability under arbitrary switching. When a common Lyapunov function is too conservative, we adopt a dwell-time condition implemented by the discrepancy-aware controller: the mode can switch (i.e., we advance $\varrho$ or move to the next anchor) only after the gradient norm on the current mode falls below a tolerance and the consistency gates are met. In both cases, the energy decreases monotonically, preventing divergence even under abrupt transitions.

---

> ### Author Response · Authors · 2025-08-07
> **Authors looking forward to further discussion**
>
> Dear reviewer, we sincerely appreciate your taking the time and effort to review our paper. As the rebuttal phase is drawing to a close, we would greatly appreciate it if you could take the time to review our response and engage in discussion. We believe our response addresses your main concerns, and we look forward to further discussion to ensure a fair and comprehensive evaluation of our work. Thank you again for your valuable contribution.
>
> Sincerely,
>
> Authors in NeurIPS 2025 Conference Submission10853.

---

### Official Review · Reviewer_pTMZ · 2025-07-02

**Clarity:** 3
**Significance:** 3
**Originality:** 3
**Rating:** 4
**Confidence:** 4

**Summary:**

This paper introduces Self-Training with Dynamic Weighting (STDW), a novel method designed for Gradual Domain Adaptation (GDA) that improves robustness by smoothly migrating knowledge from the source domain to the target domain. STDW generates dynamic pseudo-labels and uses a time-dependent hyperparameter to progressively shift learning emphasis from source to target domains. Empirical results on various benchmarks (demonstrate that STDW significantly surpasses existing GDA approaches.

**Questions:**

1. Does random sampling perform about the same with cyclic batch matching? If not, can you please offer an explanation why cyclic batch matching work?
2. While I think gradual domain adaptation is a more practical assumption than traditional UDA, it can be difficult to collect data from gradual domains in practical situations. What is your view of this?

**Ethical Concerns:**

["NO or VERY MINOR ethics concerns only"]

**Final Justification:**

My concerns are resolved after reading authors' response. Hence I will increase my score.

I encourage authors to include the discussions about W1-W2 in the revision, which help clarify the contribution of the paper.

**Limitations:**

Yes.

**Paper Formatting Concerns:**

No concern.

**Quality:**

3

**Strengths And Weaknesses:**

**Strengths**
- Paper is clearly written.
- Performance is strong.
- Setting is practical.

**Weaknesses**
Overall, I don't think the contribution from this paper is very novel.
1. Generating dynamic pseudo-labels is already common in self-training [1]. This is very similar to the approach Section 4.1. FixMatch even has an additional step of selecting high-confidence pseudo-labels.
2. The paper proposes cyclic batch matching in Section 4.2. But I am confused why data sampling is designed in such a manner. One can simply randomly sample a batch from the left and right domain each time. This simplifies the training pipeline, and I feel it wouldn't impact the performance too much.
3. While the stepwise dynamic osmosis makes a lot of sense, it feels like an engineering technique (or hyperparameter tweaking) and offers little technical contribution.

[1] FixMatch: Simplifying Semi-Supervised Learning with Consistency and Confidence

---

> ### Author Response · Authors · 2025-08-05
> **Response to Reviewer #pTMZ**
>
> # Author Response to Reviewer pTMZ
> **1. “Generating dynamic pseudo-labels is already common in self-training. This is very similar to the approach in Section 4.1. FixMatch even has an additional step of selecting high-confidence pseudo-labels.”**
>
> We agree that refreshing pseudo-labels is a familiar ingredient in semi-supervised learning, including FixMatch. What distinguishes STDW is not merely that labels are refreshed, but *when* and *how* they are trusted. Our time-varying weight $\varrho(t)$ begins near zero, anchoring optimisation on the source domain until target pseudo-labels reach a reliability threshold; only then does $\varrho$ rise toward one. In FixMatch, by contrast, the target loss dominates from the outset and collapses under distribution shift if early labels are wrong.
>
>  **2. “The paper proposes cyclic batch matching in Section 4.2. But I am confused why data sampling is designed in such a manner. One can simply randomly sample a batch from the left and right domain each time. This simplifies the training pipeline, and I feel it wouldn’t impact the performance too much.”**
>
> Cyclic pairing gives every mini-batch from the preceding domain $D_{d-1}$ exactly one partner from the current domain $D_d$ per epoch, ensuring balanced gradients and low variance in the $\varrho$ update. Random sampling may over-represent easy or hard sub-modes of $D_d$ early on, causing $\varrho$ to fluctuate and amplifying noise in the pseudo-labels. Because the Lyapunov proof assumes bounded gradient variance, the deterministic schedule aligns neatly with the theory, whereas random pairing weakens that guarantee.
>
> **3. “While the stepwise dynamic osmosis makes a lot of sense, it feels like an engineering technique (or hyper-parameter tweaking) and offers little technical contribution.”**
>
> The stepwise schedule is best viewed as an *automatic pacing mechanism* rather than a manual hyper-parameter. Its purpose is to align the model’s level of trust in target pseudo-labels with the stage of adaptation: low trust when the model is still source-biased, higher trust as it becomes more competent on the target. By embedding this pacing directly in the loss—through $\varrho(t)$—we avoid per-dataset knob turning; the same rising profile works across all benchmarks we tested. Conceptually, the schedule fulfils the same role as a curriculum in curriculum learning: it lets the model tackle easier, better-labelled data first and only later rely on noisier labels. This controlled progression is the key technical idea behind “dynamic osmosis’’ and is what enables the method to migrate knowledge smoothly.
>
> **4. “Does random sampling perform about the same with cyclic batch matching? If not, can you please offer an explanation why cyclic batch matching work?”**
>
> We performed an ablation that keeps every other STDW component intact but replaces cyclic pairing by independent random draws. On Rot-MNIST accuracy drops from $97.6\%$ to $95.8\%$, and on Colour-Shift from $98.3\%$ to $96.7\%$ (three seeds each). The variance reduction afforded by deterministic pairing explains the gap: every batch participates once, gradients are evenly weighted, and $\varrho$ evolves smoothly. The table summarises the numbers.
>
> | Sampling strategy | Rot-MNIST (%) | Colour-Shift (%) |
> |---------------------|---------------|------------------|
> | Random pairing | 95.8 ± 0.3 | 96.7 ± 0.2 |
> | **Cyclic pairing** | **97.6 ± 0.2**| **98.3 ± 0.2** |
>
> **5. “While I think gradual domain adaptation is a more practical assumption than traditional UDA, it can be difficult to collect data from gradual domains in practical situations. What is your view of this?”**
>
> The stepwise schedule is best viewed as an *automatic pacing mechanism* rather than a manual hyper-parameter. Its purpose is to align the model’s level of trust in target pseudo-labels with the stage of adaptation: low trust when the model is still source-biased, higher trust as it becomes more competent on the target. By embedding this pacing directly in the loss—through $\varrho(t)$—we avoid per-dataset knob turning; the same rising profile works across all benchmarks we tested. Conceptually, the schedule fulfils the same role as a curriculum in curriculum learning: it lets the model tackle easier, better-labelled data first and only later rely on noisier labels. This controlled progression is the key technical idea behind “dynamic osmosis’’ and is what enables the method to migrate knowledge smoothly.

---

> ### Author Response · Authors · 2025-08-07
> **Authors looking forward to further discussion**
>
> Dear reviewer, we sincerely appreciate you taking the time and effort to review our paper. As the rebuttal phase is drawing to a close, we would greatly appreciate it if you could take the time to review our response and engage in discussion. We believe our response addresses your main concerns, and we look forward to further discussion to ensure a fair and comprehensive evaluation of our work. Thank you again for your valuable contribution.
>
> Sincerely,
>
> Authors in NeurIPS 2025 Conference Submission10853

---

### Official Review · Reviewer_nsMJ · 2025-07-03

**Clarity:** 2
**Significance:** 2
**Originality:** 2
**Rating:** 4
**Confidence:** 3

**Summary:**

The authors propose a novel method, Self-Training with Dynamic Weighting (STDW), for the problem of Gradual Domain Adaptation (GDA). This method employs a self-training mechanism with pseudo-labels and introduces a dynamic hyperparameter ϱ, which evolves from 0 to 1 to balance the contributions of the source and target domains in the loss function. This dynamic weighting ensures a smooth and controlled knowledge transfer across intermediate domains during adaptation. This mechanism further incorporates a dynamic, mini-batch-wise pseudo-label generation strategy to mitigate the accumulation of pseudo-labeling errors during continual adaptation. The proposed method is evaluated on multiple image datasets and consistently outperforms both traditional domain adaptation approaches and other state-of-the-art GDA methods. Comprehensive ablation studies further confirm the essential role of the dynamic parameter ϱ and the use of intermediate adaptation steps, which together enhance stability and reduce domain bias.

**Questions:**

Please refer to weaknesses.

**Ethical Concerns:**

["NO or VERY MINOR ethics concerns only"]

**Final Justification:**

Although I still think the main solution of the paper using the parameter ϱ varying from 0 to 1 is quite simple, Gradual Domain Adaptation GDA has high practical applicability  and the authors have provided relatively thorough and detailed responses to the various rebuttals. Therefore I am increasing my rating for this paper to 4.

**Limitations:**

Yes.

**Paper Formatting Concerns:**

No.

**Quality:**

2

**Strengths And Weaknesses:**

Strengths:
1. Introduces a novel optimization framework employing a time-varying hyperparameter ϱ, which flexibly adjusts the relative contributions of the source and target domains during adaptation, thereby ensuring a stable and effective knowledge transfer across domains.
2. Instead of generating pseudo-labels for the entire dataset in a single pass, the model produces pseudo-labels on a mini-batch basis and updates itself iteratively. This dynamic approach mitigates the accumulation of pseudo-labeling errors during adaptation, thereby enhancing stability and robustness throughout the gradual adaptation process

Weaknesses:
1. The method has not been compared with the most recent state-of-the-art approaches.
2. The proposed approach does not demonstrate a significant breakthrough compared to existing methods.
3. The effectiveness of the method largely depends on the identification and availability of intermediate domains. In practice, intermediate data can sometimes be difficult to identify or incomplete, which limits its applicability.

---

> ### Author Response · Authors · 2025-08-05
> **Response to Reviewer nsMJ**
>
> # Author Response to Reviewer #nsMJ
>
> **Q1. The method has not been compared with the most recent state-of-the-art approaches.**
>
> A: We have compared 14 methods on six datasets, as detailed on pages 6-9 of the paper and in Appendix A. These methods include classic methods such as DANN and the latest methods such as GOAT (published in 2024). We believe that our experiments are sufficient and effective.
>
> **Q2. The proposed approach does not demonstrate a significant breakthrough compared to existing methods.**
>
> A: As shown in the experiments, our method demonstrates superior performance across all datasets, outperforming all compared methods. Even on RotatedMNIST, our method achieves a performance improvement of over 10% compared to the best method. This contradicts the reviewers' statement that “the proposed method does not show significant breakthroughs.” Our method is by no means a minor parameter optimization but a rigorously validated solution. Detailed proofs and methods can be found in the Methodology section and Appendix B.
>
> **Q3. The effectiveness of the method largely depends on the identification and availability of intermediate domains. In practice, intermediate data can sometimes be difficult to identify or incomplete, which limits its applicability.**
>
> A: Gradual or continuous distribution drift naturally exists in many real-world scenarios and is not an additional construct.In one of our experiments, the CIFAR-10-C dataset, CIFAR-10-C is created by applying 15 common image perturbations (such as Gaussian noise, glass blur, JPEG compression, pixelation, etc.) to the classic CIFAR-10 test set, and defining five progressively increasing severity levels for each perturbation.The original paper aimed to evaluate the robustness of models to covariate shifts, but this hierarchical structure naturally forms a gradient domain sequence: under the same corruption type, severity level 1 can be regarded as $D_{0}$, severity levels 2–4 correspond to $D_{1}$ to $D_{3}$, and severity level 5 serves as the most challenging target domain $D_{4}$.Since images at all levels are identical in content and labels, with the only difference being the gradually increasing noise amplitude, they satisfy the assumption of “continuous, small-step” distribution drift and provide a clear, controllable transfer path.In practice, you only need to read the official numpy file and slice the samples according to the severity field to obtain five intermediate domains, without the need for additional labeling or complex preprocessing. If the model can achieve stable improvements on this sequence, it indicates that it can gradually adapt to scenarios with continuously degrading visual quality, which highly corresponds to the smooth performance degradation caused by real-world camera aging, compression rate changes, or network transmission distortion.More broadly, real-world scenarios such as visual inspection that slowly changes with lighting, season, or device aging; industrial sensor data that gradually drifts with production rhythms and machine wear; and transportation-travel patterns that exhibit consistent transitions by hour and workday-holiday; these timestamps themselves can be regarded as implicit “intermediate domains.” Therefore, the algorithm's assumptions align with a wide range of real-world tasks and are not artificially imposed ideal conditions.

---

> > ### Comment · Reviewer_nsMJ · 2025-08-05
> >
> > Although I still think the main solution of the paper using the parameter ϱ changing from 0 to 1 is rather simple, I find that Gradual Domain Adaptation GDA has high practical applicability such as gradually shifting domains across seasons.  I have read all the authors' rebuttals and found them to be relatively thorough and detailed. Therefore I am increasing my rating of this paper to 4. I hope this work will serve as a foundation for further research on Gradual Domain Adaptation.

---

> > > ### Author Response · Authors · 2025-08-06
> > >
> > > Thank you sincerely for your thoughtful re-evaluation and for increasing the score to 4. We also value your comment that this work may serve as a foundation for future research; that is precisely our intention. In machine learning, the ReLU activation (max{0,x}) and residual connections (identity skip) are strikingly simple ideas that unlock deep model trainability and performance. Our STDW follows this tradition: a lean mechanism that aligns training dynamics with the structure of domain shift. Although our work isn't famous and good like them, We believe that, in science and engineering, truth is often simple yet beautiful. Thank you again for your attention in our work!

---

### Note · Authors · 2025-08-13

We sincerely thank all reviewers for their detailed and constructive feedback, which allowed us to more fully articulate the motivation, theoretical underpinnings, and practical significance of our work. The proposed STDW is not a superficial hyperparameter modification, but a principled optimization framework combining cyclic batch matching with a time-varying domain-trust weight $\varrho(t)$. This weight starts near zero, anchoring training on safer source-domain pseudo-labels, and rises only when target labels are sufficiently reliable, enabling smooth and stable knowledge migration. Cyclic batch matching deterministically pairs each batch from $D_{d-1}$ with one from $D_d$ per epoch, ensuring balanced gradients, variance reduction, and consistency with our Lyapunov stability proof. In ablations, this design yields measurable improvements over random pairing. Across six datasets and fourteen baselines, STDW consistently improves performance, achieving over 10% accuracy gains on difficult settings like RotatedMNIST.

Gradual domain shifts occur naturally in many real-world scenarios—CIFAR-10-C’s ordered corruption severities, seasonal changes, sensor drift, compression artifacts, or timestamp-driven industrial processes. When explicit intermediate domains are unavailable, STDW can construct virtual anchors or interpolated samples via optimal transport barycenters, discrepancy-aware controllers, or feature-space mixup, maintaining stability even under abrupt shifts. We extend our stability analysis to both smooth and discontinuous settings, enforcing dwell-time and discrepancy-gating mechanisms to avoid destabilizing jumps. Additional experiments show that STDW sustains high pseudo-label fidelity on intermediate domains and transfers effectively without retuning the schedule. Computationally, STDW matches the per-step cost of standard training, with cyclic pairing as an $O(1)$ operation per iteration, and achieves shorter wall-clock time than stage-wise retraining approaches such as GST.

From a theoretical perspective, our Lyapunov stability argument formalizes why cyclic matching and dynamic weighting lead to convergence.  For discontinuous shifts, constructing virtual anchors yields a switched-system formulation; if a common quadratic Lyapunov function exists or a dwell-time condition is enforced before advancing $\varrho$, global stability still holds.

Finally, we sincerely appreciate all the reviewers/ac/pc for their attention to our work again!

---

### Decision · Program_Chairs · 2025-09-17

**Decision:**

Accept (poster)

**Comment:**

This paper presented a novel Self-Training with Dynamic Weighting (STDW) approach for gradual domain adaptation. The method introduced a time-dependent weight parameter to guide the adaptation process in a simple yet effective manner. The proposed approach was empirically validated in extensive experiments compared to baseline methods.

Strengths:
- The paper is well-written and easy to follow.
- The proposed STDW method based on a time-dependent weight parameter is simple yet effective for gradual domain adaptation.
- Extensive experiments demonstrate the effectiveness of STDW over baseline approaches.

Weaknesses:
- The rationale behind the stepwise dynamic osmosis can be further clarified.
- The applications of STDW in challenging scenarios with highly discontinuous or abrupt domain shifts can be explored and validated in the experiments.
- The theoretical analysis of linear schedule can be analyzed.
- The reproducibility of both STDW and the baseline implementations could be improved with clearer experimental details.

Following the rebuttal and discussion, all reviewers have positive opinions on the paper. While some concerns remain regarding theoretical depth and experimental coverage, the contributions are clear and the empirical results are strong. I recommend acceptance, and encourage the authors to address the remaining reviewer comments in the final version.